# LATENT SPACE SYMMETRY DISCOVERY

## ABSTRACT

Equivariant neural networks require explicit knowledge of the symmetry group. Automatic symmetry discovery methods aim to relax this constraint and learn invariance and equivariance from data. However, existing symmetry discovery methods are limited to linear symmetries in their search space and cannot handle the complexity of symmetries in real-world, often high-dimensional data. We propose a novel generative model, Latent LieGAN (`LaLiGAN`), which can discover nonlinear symmetries from data. It learns a mapping from data to a latent space where the symmetries become linear and simultaneously discovers symmetries in the latent space. Theoretically, we show that our method can express any nonlinear symmetry under certain conditions. Experimentally, our method can capture the intrinsic symmetry in high-dimensional observations, which results in a well-structured latent space that is useful for other downstream tasks. We demonstrate the use cases for `LaLiGAN` in improving equation discovery and long-term forecasting for various dynamical systems.

## 1 INTRODUCTION

Symmetry plays an important role in the success of deep neural networks (Bronstein et al., 2021). Many equivariant networks have been developed to enforce various symmetries in data from images to graphs (Weiler & Cesa, 2019; Cohen et al., 2019a; Zaheer et al., 2017; Finzi et al., 2020; Kondor & Trivedi, 2018; Cohen et al., 2019b; Finzi et al., 2021; Bekkers, 2019). A critical limitation of existing equivariant networks is that they require knowing the symmetry a priori.

However, for complex real-world data, the underlying symmetries may be unknown or challenging to articulate through programming. Recent years have seen exciting attempts towards automatic symmetry discovery from data (Dehmamy et al., 2021; Moskalev et al., 2022; Benton et al., 2020; Zhou et al., 2021), but most of them search in only a limited space of symmetries, such as subsets of known groups or finite groups. LieGAN (Yang et al., 2023) can discover various types of symmetries, but its search space is still constrained to general linear groups. Successful discovery can only be achieved when observations are measured in an ideal coordinate system where linear symmetry is present. Unfortunately, real-world data often contain nonlinear symmetries, such as high-dimensional dynamics that evolve on a low-dimensional manifold (Champion et al., 2019), or 2D images of 3D objects (Garrido et al., 2023).

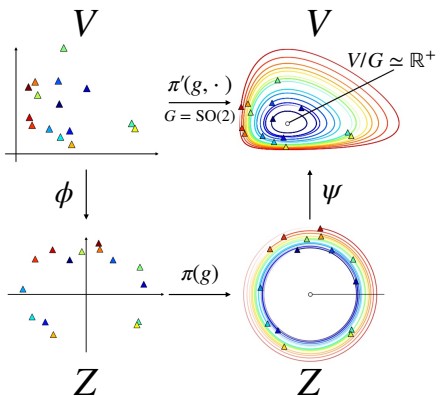

Figure 1: An example of SO(2) nonlinear group action $\pi'$ on $V = \mathbb{R}^2$ and its decomposition into an encoder $\phi$, a linear representation $\pi$ and a decoder $\psi$. Each trajectory is a group action orbit containing a random $v \in V$.

Another line of study focuses on learning equivariant representations (Park et al., 2022; Yu et al., 2022; Dangovski et al., 2021; Quessard et al., 2020). These approaches learn a latent embedding space with particular symmetries. However, they still require prior knowledge about the symmetry in the latent space. Also, they often assume additional information about group transformation associated with each data point, which is not always available in practice.

In this work, we propose a novel framework, `LaLiGAN`, for discovering symmetries of *nonlinear* group actions. `LaLiGAN` decomposes the group transformations into nonlinear mappings between data space and latent space, and a linear group representation in the latent space. Figure 1 provides an example of such decomposition, where a nonlinear action of $SO(2)$ on $V = \mathbb{R}^2$ corresponds to standard 2D rotation on latent vectors $z = \phi(v)$. Then, we utilize an existing symmetry discovery algorithm (Yang et al., 2023) with careful adaptations for discovering symmetries in the latent space. Normally, our framework has learnable group representation and does not require information about specific groups. However, when the symmetry group is known, it can also be used to learn equivariant representations without the information of group elements associated with each data sample. It is a highly flexible framework and can be applied to scenarios with scarce domain knowledge.

The significance of latent space symmetry discovery is multi-fold. From the perspective of symmetry discovery, it further expands the search space of symmetries beyond linear group actions. For representation learning, learning a latent space in which symmetry becomes linear places a strong inductive bias on the structure of latent representations. Such a simple latent structure proves to be useful in various downstream tasks, such as equation discovery and long-term forecasting in temporal systems. Furthermore, compared to equivariant representation learning, as the symmetry is no longer fixed but learnable, our method can discover latent spaces with previously unknown symmetries.

In summary, our main contributions include:

- We develop `LaLiGAN`, a novel framework for discovering symmetries of nonlinear group actions.
- We provide the theoretical guarantee that `LaLiGAN` has the expressive power to approximate any nonlinear symmetry under certain conditions.
- Our method can discover well-structured latent spaces with interpretable symmetries in high-dimensional and nonlinear dynamical systems.
- The discovered symmetry can be applied to equation discovery, leading to simpler equation forms and improved long-term prediction accuracy.

## 2 RELATED WORKS

**Automatic symmetry discovery.** Automatic symmetry discovery aims to search and identify unknown symmetries in data. Current symmetry discovery techniques vary a lot in their search space for symmetries, such as learning discrete finite groups (Zhou et al., 2021; Karjol et al., 2023), learning group subsets that represent the extent of symmetry within known groups (Benton et al., 2020; Romero & Lohit, 2022; Chatzipantazis et al., 2021), and learning individual symmetry transformations on dataset distribution (Desai et al., 2022). Attempts have been made to discover general continuous symmetries based on Lie theory. For example, L-conv (Dehmamy et al., 2021) works with Lie algebra to approximate any group equivariant functions. LieGG (Moskalev et al., 2022) extracts symmetry from a learned network from its polarization matrix. LieGAN (Yang et al., 2023) proposes a general framework for discovering the symmetries of continuous Lie groups and discrete subgroups. These methods address general linear group symmetry in the data, which is the largest search space so far. Our work expands the search space to non-linear symmetries.

**Learning equivariant representation.** Instead of working in the data space where symmetry transformations can be complicated, many works use autoencoders to learn a latent space with pre-specified symmetries (Hinton et al., 2011; Falorsi et al., 2018). Among recent works, Yu et al. (2022); Park et al. (2022) learn equivariant features that can be used for downstream prediction tasks. Shakerinava et al. (2022); Dangovski et al. (2021) use contrastive losses to learn equivariant representations in a self-supervised manner. Quessard et al. (2020); Marchetti et al. (2023) focus on learning disentangled representations that are highly interpretable. Winter et al. (2022); Wieser et al. (2020) split the latent space into group-invariant and equivariant subspaces. While the emphases of these works vary, the common assumption is that *we have to know the symmetry group a priori*. Many of them also assume additional information such as group element associated with each data point (Garrido et al., 2023) or paired samples under certain transformations (Shakerinava et al., 2022). Our goal is more ambitious: design a model to simultaneously learn symmetries and the corresponding equivariant representations in latent space with minimal supervision.

**Discovering governing equations.** Latent space discovery of governing equations is first introduced in SINDy Autoencoder (Champion et al., 2019), which combines the sparse regression technique for discovering dynamics in Brunton et al. (2016) and an autoencoder network to explore coordinate transformations that lead to parsimonious equations. Several variants of this method have been developed to improve accuracy and robustness to noise (Kaheman et al., 2020; Messenger & Bortz, 2021; Fasel et al., 2022). However, due to the absence of physical constraints, their discovered equations may not respect some physical properties such as isotropy and energy conservation. We highlight this field as an important application of our symmetry discovery method, where enforcing symmetry can regularize the latent space and improve the performance of equation discovery models.

## 3 Representation vs Nonlinear Group Action

Equivariant neural networks build on the notion of symmetry groups and their transformations on data. Given a vector space $V$, a group $G$ transforms $v \in V$ via a group action $\pi : G \times V \to V$ which maps the identity element $e$ to identity transformation, i.e. $\pi(e, v) = v$, and is compatible with group element composition, i.e. $\pi(g_1, \pi(g_2, v)) = \pi(g_1 g_2, v)$.

Many existing equivariant networks assume that the group acts linearly on the input vector space. Examples include E(2) symmetry acting on planar image signals Weiler & Cesa (2019), and SO(3) symmetry acting on spherical signals Cohen et al. (2018). In these cases, the linear group action is called a group representation. The group representation is defined as a map $\rho : G \to \text{GL}(n)$ where $\rho(g) \in \mathbb{R}^{n \times n}$ is an invertible matrix that transforms any vector $v \in R^n$ by matrix multiplication. Given the group representations on the input and the output spaces, a $G$-equivariant network $f : X \to Y$ needs to satisfy $\rho_Y(g) f(x) = f(\rho_X(g) x)$. A special case of equivariance is invariance, where the group action on the output space is trivial, i.e. $\rho_Y(g) = \text{id}$.

Equivariant networks with such linear symmetry transformations have several limitations. It is not always possible to find a linear action of the group on the data, e.g. the action of SO(3) on 2D images of 3D objects. Also, we may not even know the symmetry group $G$ itself, so learning equivariant representations for known groups is also not an option.

Our goal is to discover both the **symmetry group** and its **nonlinear group action** on the data. Concretely, given the input and output data space $X \subseteq \mathbb{R}^n$, $Y \subseteq \mathbb{R}^m$, and the data samples $(x_i, y_i) \in X \times Y$ with an underlying function $y = f(x)$, we want to find a group $G$ and its nonlinear actions $\pi'_X : G \times X \to X$ and $\pi'_Y : G \times Y \to Y$ such that $\pi'_Y(g, f(x)) = f(\pi'_X(g, x))$. We denote nonlinear group actions as $\pi'$ to distinguish them from group representations. In the following sections, we will also refer to group representations and nonlinear group actions as linear symmetries and nonlinear symmetries.

We will use the theory of Lie groups to describe the continuous symmetry groups of data. We provide some preliminaries about Lie groups and their representations in Appendix B.

## 4 LaLiGAN: Discovering Nonlinear Symmetry Transformations

### 4.1 Decomposing the Nonlinear Group Action

Our major goal is to learn a nonlinear action of a group $G$ on a vector space $V$: $\pi' : G \times V \to V$. While we can use a neural network $f_\theta$ to directly approximate this function, it does not guarantee the identity and compatibility conditions for a proper group action, i.e. $f_\theta(\text{id}, x) = x$ and $f_\theta(g_1, f_\theta(g_2, x)) = f_\theta(g_1 g_2, x)$. Instead, we propose to decompose the nonlinear group action as nonlinear maps and a linear group representation. Concretely, we represent any nonlinear group action $\pi' : G \times V \to V$ as

$$\pi'(g, \cdot) = \psi \circ \pi(g) \circ \phi, \tag{1}$$

where $\phi : V \to Z$ and $\psi : Z \to V$ are functions parametrized by neural networks, and $\pi(g) : G \to \text{GL}(k)$ is a group representation acting on the latent vector space $Z = \mathbb{R}^k$. We specify the dimensionality of $Z$ as a hyperparameter based on specific tasks. One can easily verify that

**Proposition 4.1.** *If $\phi$ and $\psi$ are inverse of each other, then $\pi'(g, \cdot) = \psi \circ \pi(g) \circ \phi$ is a valid group action that satisfies identity and compatibility axioms.*

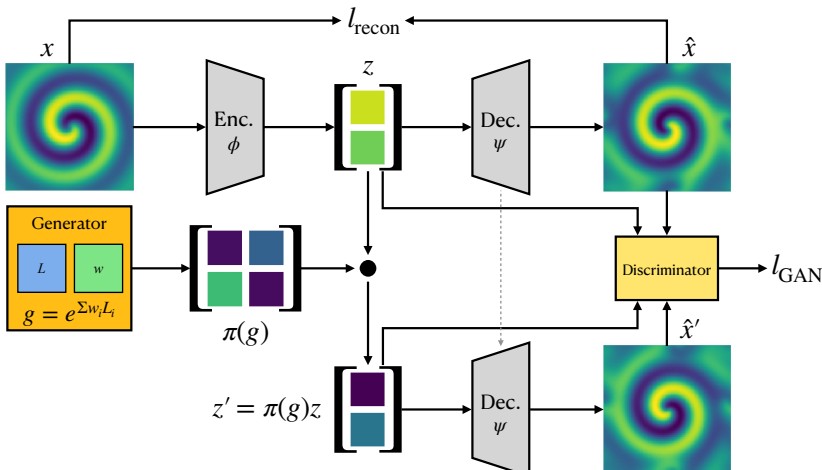

Figure 2: Overview of the proposed `LaLiGAN` framework. The encoder maps the original observations to a latent space. The latent representation is transformed with the linear group action from the generator. The decoder reconstructs the inputs from original and transformed representations. The discriminator is trained to recognize the difference between the original and the transformed samples.

In practice, we train the networks $\phi$ and $\psi$ with a reconstruction loss $l_{\text{recon}} = \mathbb{E}_v \|\psi(\phi(v)) - v\|^2$ to ensure they are the approximate inverse of each other. Intuitively, $\phi$ and $\psi$ form an autoencoder that maps between the input vector space and a latent space. Through the decomposition of the nonlinear group action, our method learns (1) the symmetry group on a latent space via its linear representation, and (2) a pair of inverse mappings between the input space and the symmetric latent space.

We can provide theoretical guarantees for the expressivity of such a decomposition. The following theorem shows that our proposed decomposition and neural network parametrization can approximate nonlinear group actions under certain conditions. Detailed proof is deferred to Appendix C.

**Theorem 4.2** (Universal Approximation of Nonlinear Group Action). *Let $G \leq \text{GL}(k; \mathbb{R})$ be a compact Lie group that acts smoothly, freely and properly on $V = \mathbb{R}^k$ via a continuous group action $\pi' : G \times V \to V$. The group action, restricted to any bounded subset of the group, can be approximated by the decomposition $\pi'(g, \cdot) \approx \psi \circ \pi(g) \circ \phi$ if it admits a simply connected orbit space $V/G$, where $\psi$ and $\phi$ are fixed arbitrary-width neural networks with one hidden layer, and $\pi$ is a linear group representation.*

## 4.2 SYMMETRY DISCOVERY

Now that we have constructed the nonlinear group action, we proceed to discover the symmetry group $G$. We restrict our search space to $G \leq \text{GL}(k)$, where $k$ is the latent dimensionality defined in the previous decomposition. In this way, we can represent any group element $g$ by its standard representation $\pi(g) \in \mathbb{R}^{k \times k}$. We expect this search space of the general linear group to be big enough to cover the types of symmetries in most real-world systems.

We follow the approach in Yang et al. (2023) to discover the linear symmetry with generative adversarial training. Concretely, a symmetry generator learns a Lie algebra basis $\{L_i \in \mathbb{R}^{k \times k}\}$ and generates the standard representations of group elements by sampling the linear combination coefficients $w_i \in \mathbb{R}$ for the Lie algebra basis:

$$w_i \sim \gamma(w), \ \pi(g) = \exp\left[\sum_i w_i L_i\right] \tag{2}$$

where $\gamma$ is a distribution (e.g. Gaussian) for the coefficients and $\exp$ denotes the matrix exponential. As the Lie algebra basis $\{L_i\}$ uniquely determines the structure of the Lie group, we can learn the symmetry group by learning these $L_i$ via standard gradient-based optimization techniques.

Then, the symmetry generator introduced in (2) samples random group elements that transform the data points $v_i = (x_i, y_i)$. The discriminator is trained to distinguish the original "real" data and the

transformed "fake" data. The generator and the discriminator are trained adversarially so that the generator learns to produce group elements that preserve the data distribution while transforming each data point. The group learned by the generator is then considered the discovered symmetry of the data. Figure 2 shows the overall pipeline of our method.

We term our method Latent LieGAN (`LaLiGAN`), as we learn the Lie group representations on a latent space. A key difference of our method is the nonlinearity of the group action on data, which is achieved through the decomposition in (10). Besides, we use the latent representations as the discriminator input. The latent vectors before the group transformations are the "real" samples, and those after the transformations are "fake". Optionally, we also concatenate each latent vector with its reconstruction in observation space as the discriminator input, which is shown to accelerate convergence. In the most general form, our training objective is formulated as

$$l_{\text{total}} = w_{\text{GAN}} \cdot l_{\text{GAN}} + w_{\text{recon}} \cdot l_{\text{recon}}, \quad l_{\text{recon}} = \mathbb{E}_v \|(\psi \circ \phi)(v) - v\|^2,$$

$$l_{\text{GAN}} = \mathbb{E}_{v,g}\Big[ \log D(\phi(v), (\psi \circ \phi)(v)) + \log(1 - D((\pi(g) \circ \phi)(v), (\psi \circ \pi(g) \circ \phi)(v))) \Big] \quad (3)$$

where $D$ is the discriminator, $\pi(g) = \exp(w^i L_i)$ is the representation of the group element sampled from the generator, and $\phi$ and $\psi$ are neural networks that compose the nonlinear group action together with $\pi(g)$. The learnable components include $D$, $L_i$, $\phi$ and $\psi$, which are optimized under the joint objective $l_{\text{total}}$. The loss weighting coefficients $w_{\text{GAN}}$ and $w_{\text{recon}}$ are selected based on specific tasks.

### 4.3 STRUCTURING THE LATENT SPACE

**Disentangled representation.** Latent space representations may capture different aspects of the observations. Consider an image of $N$ 3D objects as an example. A possible latent representation consists of the orientation of each object $r_o \in \mathbb{R}^{3N}$, the camera perspective $r_c \in \mathbb{R}^3$, light intensity $i \in \mathbb{R}^+$, etc. Each component can be transformed by a separate group action, independent of each other. For these scenarios, we provide the option to specify how the latent space is decomposed as independent subspaces, i.e. $Z = \oplus_{i=1}^N Z_i$, each of which is acted on by a symmetry group $G_i$. This avoids searching in the unnecessarily large space of group actions with no nontrivial invariant subspace. This aligns with the notion of disentangled representation in Higgins et al. (2018).

**Regularizing the latent structure.** The latent space produced by an encoder network can be largely arbitrary, leading to fallacious symmetry or no symmetry at all. We observe some failure modes caused by undesirable latent space structures and propose some regularization methods. First, the latent representations tend to collapse to a low-dimensional subspace where nontrivially parametrized group representations can act as identity. Such a fallacious symmetry provides an easy workaround for the symmetry generator. For example, this happens in Figure 3a, where the transformations generated by $L = [2, -2; -1, 1] \in \mathbb{R}^{2\times2}$ leave the latent representations in a 1D subspace approximately unchanged. This is undesirable because we want the symmetry generator to learn nontrivial transformations. In practice, we use orthogonal parametrization in the final linear layer of the encoder to enforce a different output in each dimension.

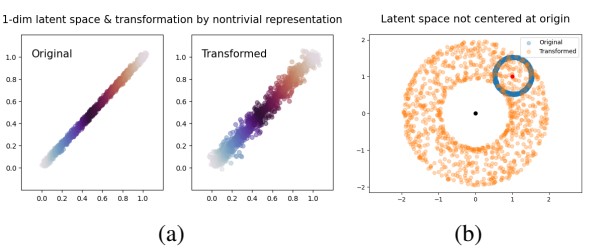

Figure 3: Potential failure modes in latent space symmetry discovery. (a) Fallacious symmetry in low-dimensional subspace. (b) Absence of symmetry in a biased latent space.

Another failure mode occurs when the latent representations are not centered at the origin. The linear group representation $v \mapsto \pi(g)v$ implicitly assumes that the vector space is centered at the origin and cannot describe the symmetry otherwise. Figure 3b provides an example of a circular latent space centered at $(1, 1)$. Directly applying the SO(2) transformations result in a different distribution. We observe that the encoder struggles to learn to center the latent representations at the origin. Therefore, we enforce this property by normalizing each batch of data to have zero mean before applying the transformations from the symmetry generator.

## 4.4 APPLICATIONS OF LATENT SYMMETRY DISCOVERY

**Learning equivariant representation.**   Learning equivariant representation can be viewed as a special case of our method, where the symmetry group $G$ and its representation $\pi$ are known. Our encoder $\phi$ then becomes a $G$-equivariant function in the sense that

$$\phi(\pi'(g,x)) = \phi((\psi \circ \pi(g) \circ \phi)(x)) = \pi(g)\phi(x) \tag{4}$$

In other words, by fixing $\pi$ to a known group representation, our method learns a $G$-equivariant representation $z = \phi(x)$. Compared to other methods, `LaLiGAN` can learn equivariant representation without any knowledge of the group transformation associated with each data sample.

**Joint discovery of governing equation.**   `LaLiGAN` is analogous to latent space equation discovery techniques (Champion et al., 2019) in terms of using an autoencoder network for nonlinear coordinate transformations. We can use the latent space learned by `LaLiGAN` for discovering equations. Concretely, if we want to find a latent space governing equation parameterized by $\theta$: $\dot{z} = F_\theta(z)$, where $z = \phi(x)$ is obtained from our encoder network, we fix the encoder $\phi$ and optimize $\theta$ with the objective $l_{eq} = \mathbb{E}_{x,\dot{x}}\|(\nabla_x z)\dot{x} - F_\theta(z)\|^2$.

While equation discovery and symmetry discovery are two seemingly distinct tasks, we will show in the experiment that learning a symmetric latent space can significantly improve the quality of the discovered equation in terms of both its simplicity and its long-term prediction accuracy.

## 5 LATENT SYMMETRY IN DYNAMICAL SYSTEMS

### 5.1 DATASETS

**Reaction-diffusion.**   Many high-dimensional datasets in practical engineering and science problems derive from dynamical systems governed by partial differential equations. These systems often do not exhibit simple linear symmetries in the observation space, but their dynamics might evolve on a low-dimensional manifold with interesting symmetry properties. As an example, we consider a $\lambda - \omega$ reaction-diffusion system (Champion et al., 2019) governed by

$$
\begin{aligned}
u_t =& (1 - (u^2 + v^2))u + \beta(u^2 + v^2)v + d_1(u_{xx} + u_{yy}) \\
v_t =& -\beta(u^2 + v^2)u + (1 - (u^2 + v^2))v + d_2(u_{xx} + u_{yy})
\end{aligned}
\tag{5}
$$

with $d_1 = d_2 = 0.1$ and $\beta = 1$. We discretize the 2D space into a $100 \times 100$ grid, which leads to an input dimension of $10^4$. Figure 4b visualizes a few snapshots of this system. We simulate the system up to $T = 6000$ timesteps with step size $\Delta t = 0.05$.

The reaction-diffusion system is an example of low-dimensional latent symmetry in high-dimensional observations. In fact, the absence of linear symmetry is not exclusive to high-dimensional systems. We also investigate two low-dimensional dynamics, where their nonlinear evolution prevents any kind of linear symmetry, but our method can still discover meaningful symmetries in the latent space.

**Nonlinear pendulum.**   The movement of a simple pendulum can be described by $\dot{q} = p$, $\dot{p} = -\omega^2 \sin(q)$, with $\omega$ being the natural frequency and $q$ and $p$ the angular displacement and angular momentum. In our experiment, we use $\omega = 1$. We simulate $N = 200$ trajectories up to $T = 500$ timesteps with $\Delta t = 0.02$.

**Lotka-Volterra System.**   The Lotka-Volterra equations are a pair of nonlinear ODEs that characterize the dynamics of predator-prey interaction. We consider the canonical form of the equations, $\dot{p} = a - be^q$, $\dot{q} = ce^p - d$, where $p$ and $q$ are the logarithm population densities of prey and predator, and the parameters $a, b, c, d$ indicate the growth and death rate of the two populations. In our experiment, we use $a = 2/3, b = 4/3$, and $c = d = 1$. We simulate $N = 200$ trajectories up to $T = 10^4$ timesteps with $\Delta t = 0.002$.

### 5.2 SYMMETRY DISCOVERY

We train `LaLiGAN` to learn the nonlinear mappings between observations and latent representations, along with the linear symmetry in the latent space. We aim to discover the equivariance of latent

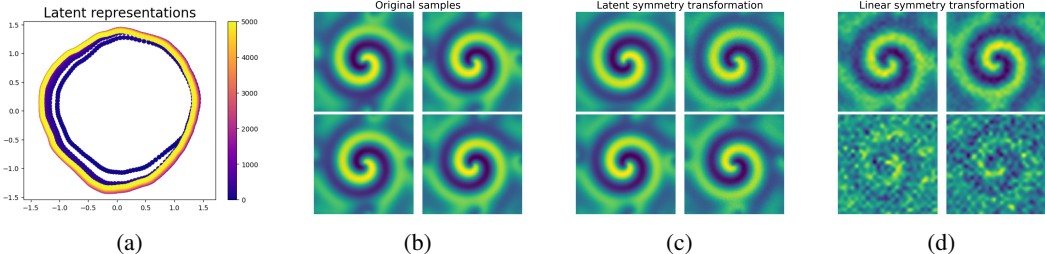

Figure 4: Symmetry discovery in reaction-diffusion system with 2D latent space. (a) Latent representations of the system at all timesteps. (b) Randomly selected samples from the dataset. (c) Samples transformed by `LaLiGAN` are similar to the original data. (d) Samples transformed by the baseline, linear LieGAN, are significantly different from the original data.

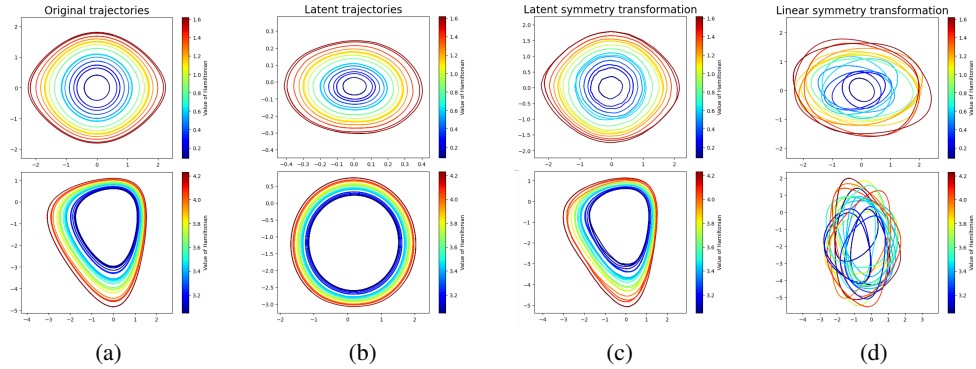

Figure 5: Latent symmetry discovery in nonlinear pendulum (upper) and Lotka-Volterra equations (lower). (a) Original trajectories of the systems. The color of each trajectory corresponds to its Hamiltonian. (b) The trajectories are mapped to a symmetric latent space. (c) Original trajectories transformed by `LaLiGAN`. (d) Original trajectories transformed by linear LieGAN.

dynamics, i.e. $z_{t+1} = f(z_t) \Rightarrow gz_{t+1} = f(gz_t)$. Therefore, we take two consecutive timesteps as input, encode them to latent representations with the same encoder weights, and apply the same transformations sampled from the symmetry generator.

For the reaction-diffusion system, we follow the setting in Champion et al. (2019) and set the latent dimension $k = 2$. Figure 4a shows how the system evolves in the latent space throughout $T = 5000$ timesteps. The Lie algebra basis discovered in the latent space is $L = [0.06, -3.07; 3.05, -0.04]$. This suggests an approximate $SO(2)$ symmetry, which is also evident from the visualization.

For the pendulum and the Lotka-Volterra system, we also set the latent dimensions to 2, which is the same as their input dimensions. Figure 5b shows the trajectories of these two systems in the latent space, with the discovered symmetries $L_{\text{pendulum}} = [0, -5.24; 2.16, 0]$ and $L_{\text{LV}} = [0, 2.43; -2.74, 0]$. These indicate rotation symmetries up to a certain scaling in the latent dimensions.

The validity of the discovered symmetry can be verified by visually inspecting the difference between the transformed and the original samples. For the reaction-diffusion system, Figure 4c shows some samples with random transformations produced by our method, which are similar to the original data displayed in Figure 4b. We also apply the original LieGAN to this task for comparison, and the transformed samples are shown in Figure 4d. These samples contain obvious artifacts and are noticeably different from the original data, which suggests the necessity of our method when linear symmetry does not exist in observation space.

Similarly, for the pendulum and the Lotka-Volterra system, we use the learned symmetries to transform each entire trajectory, as is shown in Figure 5c. Each trajectory is transformed from the original trajectory of the same color. While each individual data point is taken into a new position, the entire trajectories remain similar before and after transformation, suggesting that the discovered transformations are indeed the symmetries of these systems. In contrast, the linear symmetries learned by LieGAN do not preserve valid trajectories in the observation space, as shown in Figure 5d.

## 5.3 EFFECT OF LATENT DIMENSIONALITY

The latent dimension $k$ is a hyperparameter in our method. However, it is not always possible to choose the perfect latent dimension that matches the intrinsic dimension of the system and uncovers symmetry in latent space. To study the robustness of our method under a less ideal hyperparameter configuration, we set the latent dimension to $k = 3$ for the reaction-diffusion system and repeat the experiment. As shown in Figure 6a, the Lie algebra representation is skew-symmetric, which indicates the symmetry of rotations around a particular axis. This can be easily confirmed as all the latent representations roughly dwell on a circular 2D subspace. Although it is not the most simple representation, our method still manages to discover the rotation symmetry as in 2D latent space.

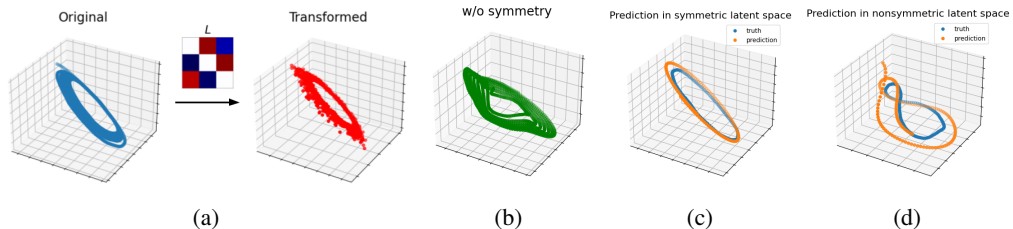

|     (a)     |     (b)     |     (c)     |     (d)     |

Figure 6: Modeling reaction-diffusion system in 3D latent space. (a) The latent representations before and after our discovered symmetry transformations. (b) The discovered latent space with SINDy but without `LaLiGAN`. (c-d) Simulation trajectory in the previous two latent spaces.

## 5.4 EQUATION DISCOVERY

We demonstrate the benefit of learning latent symmetry by using the latent space to discover governing equations. This is a commonly considered problem in these dynamical systems. We use SINDy (Brunton et al., 2016; Champion et al., 2019) as the equation discovery algorithm, with up to second

| Method | `LaLiGAN` + SINDy | SINDy AE |
|---|---|---|
| 2D | $\dot{z}_1 = \phantom{-}0.91z_2$ | $\dot{z}_1 = -0.85z_2$ |
|    | $\dot{z}_2 = -0.91z_1$ | $\dot{z}_2 = \phantom{-}0.97z_1$ |
| 3D | $\dot{z}_1 = \phantom{-}0.58z_2 - 0.40z_3$ | $\dot{z}_1 = \phantom{-}0.65z_2 - 0.16z_3 + \Theta(\mathbf{z}^2)$ |
|    | $\dot{z}_2 = -0.56z_1 + 0.54z_3$ | $\dot{z}_2 = -0.57z_1 + 0.18z_2 + \Theta(\mathbf{z}^2)$ |
|    | $\dot{z}_3 = \phantom{-}0.45z_1 - 0.57z_2$ | $\dot{z}_3 = \phantom{-}0.45z_1 - 0.57z_2 + \Theta(\mathbf{z}^2)$ |

Table 1: Equation discovery on 2D/3D latent spaces for R-D system. Complete results are available in Appendix A.1.

order polynomials as candidate functions. The comparison is made between applying SINDy on the latent space learned by our method (`LaLiGAN` + SINDy) and using the SINDy autoencoder to learn its own latent space (SINDy AE). The results for the reaction-diffusion system are shown in Table 1. The discovered equations from both methods have similar forms in the 2D latent space. In the 3D latent space, the governing equation learned in the `LaLiGAN` latent space remains linear. On the other hand, applying the SINDy autoencoder alone results in a nonsymmetric latent space (Figure 6b) and a highly complicated governing equation with second-order terms.

**Long-term forecasting.** To further verify the accuracy of the discovered equations, we use these equations to simulate the dynamics in the latent space. Concretely, given the initial input frame $x_0$, we obtain its latent representation $\hat{z}_0 = \phi(x_0)$ and predict the future $T$ timesteps by iteratively computing $\hat{z}_{t+1} = \hat{z}_t + F(\hat{z}_t) \cdot \Delta t$, where $\dot{z} = F(z)$ denotes the discovered governing equation. Then, we map the representations back to the input space by $\hat{x}_t = \psi(\hat{z}_t)$. Figure 6c and 6d show the simulated latent trajectories from the equations discovered in 3D latent space with and without `LaLiGAN`. The trajectory remains close to ground truth in the symmetric latent space but diverges quickly for the equation discovered by SINDy AE. We also evaluate the forecasting accuracy quantitatively by the relative MSE between the prediction and ground truth in the observation space, as is shown in Figure 7. Besides the symbolic models in Table 1, we also include Neural ODE

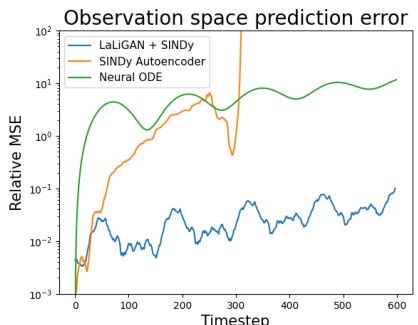

Figure 7: Relative MSE losses of long-term simulations of R-D system.

(Chen et al., 2018) as a baseline. Similar to the symbolic equation discovery, it can also predict the

dynamics at arbitrary timesteps with an ODE parametrized by neural nets. Figure 7 shows that the discovered equation learned with latent space symmetry outperforms both the equation from vanilla SINDy AE and the Neural ODE model in this task of long-term dynamics forecasting.

We also conduct the same experiments of equation discovery and long-term forecasting as for the nonlinear pendulum and the Lotka-Volterra system. While they have simple closed-form governing equations in the observation space, we find that discovering a latent space with learnable symmetry can still be beneficial. The symmetry enforces linear governing equations and leads to reduced error accumulation in long-term forecasting. The detailed results are available in Appendix A.2.

## 6 LEARNING EQUIVARIANT REPRESENTATION

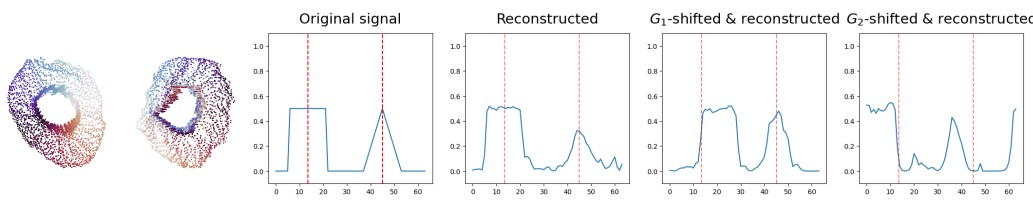

(a) Learned latent space      (b) An original signal transformed by $\text{SO}(2) \times \text{SO}(2)$ latent symmetry

Figure 8: Learning equivariant representation of the double-bump world. (a) Learned latent space as the direct sum of two 2D subspaces. The color of the data points corresponds to the location of the rectangular bump in the first component and the triangular bump in the second. (b) From left to right: (1) an original signal $x \in \mathbb{R}^{64}$; (2) reconstructed signal $\psi(\phi(x))$; (3-4) reconstructed signals from transformed latent representations, $\psi((\pi(\theta_1) \oplus I)\phi(x))$ and $\psi((I \oplus \pi(\theta_2))\phi(x))$. The red lines are the bump centers in the original signal.

When we know the linear group representation, we can use `LaLiGAN` for learning the corresponding group equivariant representation. Unlike previous works (Garrido et al., 2023; Shakerinava et al., 2022), we learn it without any knowledge of the group element associated with each data point. We consider the example of a double-bump world in Shakerinava et al. (2022). It consists of a rectangular and a triangular bump signal, both cyclically shifted in a fixed-length window. We use the original experiment setting with signal length 64 and bump length 16, visualized in Figure 8b. The cyclic translation of each bump forms an $\text{SO}(2)$ group. As each bump is shifted independently, the symmetry group for the composed signal is $\text{SO}(2) \times \text{SO}(2)$. Therefore, we use a 4-dimensional latent space $Z = \mathbb{R}^2 \oplus \mathbb{R}^2$ and fix the Lie algebra basis to $L = L_1 \oplus L_2$, $L_1 = L_2 = [0, 1; -1, 0]$.

Figure 8a shows the latent space learned by `LaLiGAN`. We observe that rotation in the first component shifts the rectangular bump, while rotation in the second component simultaneously shifts both bumps. This is also evident from the transformed and reconstructed samples in Figure 8b. This provides an example that our method can learn equivariant representations when we do not know the group transformation of each data point. We also include another experiment on $\text{SO}(3)$ equivariant representation for a 3D object in Appendix A.4.

## 7 CONCLUSION

We propose `LaLiGAN`, a novel generative modeling framework for discovering nonlinear symmetries. `LaLiGAN` decomposes the group action as a linear representation on a latent space and a pair of nonlinear mappings between the latent space and the observation space. By jointly optimizing the group representation and the nonlinear mappings, it discovers both the symmetry group and its nonlinear group action on the data. We also show that it can be applied to downstream tasks such as equation discovery, leading to equations with simpler forms and better long-term prediction accuracy.

In the future, we plan to study how the knowledge of latent space symmetry can be better incorporated into equation discovery. For example, symmetry can act as a constraint to compress the search space for equations and accelerate the search. We also plan to investigate the connection between symmetry and other physical properties such as conservation laws. Given the prevalence of symmetries in the natural world, our long-term goal is to develop a general framework for automatically discovering symmetries and other types of governing laws from data and accelerate scientific discovery process.

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

## A  SUPPLEMENTARY EXPERIMENT RESULTS

### A.1  HIGH-DIMENSIONAL REACTION-DIFFUSION SYSTEM

| Model | Discovered equation |
|---|---|
| LaLiGAN + SINDy | $\dot{z}_1 = \phantom{-}0.43z_2 - 0.53z_3$ 
 $\dot{z}_2 = -0.51z_1 + 0.66z_3$ 
 $\dot{z}_3 = \phantom{-}0.47z_1 - 0.52z_2$ |
| LaLiGAN + SINDy + PCA | $\dot{u}_1 = -0.98u_2$ 
 $\dot{u}_2 = \phantom{-}0.84u_1$ 
 $\dot{u}_3 = \phantom{-}0$ |
| SINDy AE | $\dot{z}_1 = \phantom{-}0.65z_2 - 0.16z_3 + 0.20z_1^2 + 0.11z_1z_2 + 0.29z_1z_3$ 
 $\phantom{\dot{z}_1 =}- 0.41z_2z_3 - 0.16z_3^2$ 
 $\dot{z}_2 = -0.57z_1 + 0.18z_2 - 0.24z_1z_2 + 0.46z_1z_3 - 0.18z_2^2$ 
 $\phantom{\dot{z}_2 =}- 0.26z_2z_3 + 0.29z_3^2$ 
 $\dot{z}_3 = \phantom{-}0.45z_1 - 0.57z_2 - 0.27z_1^2 + 0.18z_2^2 - 0.19z_2z_3$ |
| SINDy AE + PCA | $\dot{u}_1 = \phantom{-}0.95u_2 - 0.06u_3 + 0.09u_1u_2 + 0.16u_1u_3 - 0.59u_2u_3 - 0.12u_3^2$ 
 $\dot{u}_2 = -0.58u_1 + 0.29u_3 - 0.57u_1u_3 - 0.23u_2u_3 - 0.10u_3^2$ 
 $\dot{u}_3 = -0.06u_1^2 + 0.51u_1u_2 + 0.08u_2^2 + 0.35u_2u_3$ |

Table 2: Complete equation discovery results on 3D latent space for reaction-diffusion system.

Table 2 shows the complete results of discovering governing equations in 3D latent space for reaction-diffusion system. Applying the SINDy autoencoder alone results in a highly complicated governing equation with several second-order terms. Also, we find that the equations learned on the symmetric latent space can even be further simplified with a linear transformation introduced by principle component analysis (PCA). The $u$'s in the equations denote the principle components, sorted by the variance each component explains. In comparison, the equations from the SINDy autoencoder alone do not admit a simpler form under the linear PCA transformation.

### A.2  NONLINEAR ORDINARY DIFFERENTIAL EQUATIONS

| Method | Discovered equation |
|---|---|
| LaLiGAN + SINDy | $\dot{z}_1 = -0.94z_2$ 
 $\dot{z}_2 = \phantom{-}0.38z_1$ |
| SINDy | $\dot{z}_1 = \phantom{-}0.99z_2$ 
 $\dot{z}_2 = -0.98\sin(z_1)$ |
| SINDy AE | $\dot{z}_1 = -0.46\sin(z_2)$ 
 $\dot{z}_2 = \phantom{-}0.51z_1 + 0.42\sin(z_1)$ |

Table 3: Equation discovery for pendulum.

| Method | Discovered equation |
|---|---|
| LaLiGAN + SINDy | $\dot{z}_1 = -0.65 - 0.56z_2$ 
 $\dot{z}_2 = -0.14 + 0.67z_1$ |
| SINDy | $\dot{z}_1 = \phantom{-}0.64 - 1.28e^{z_2}$ 
 $\dot{z}_2 = -0.91 + 1.05e^{z_1}$ |
| SINDy AE | $\dot{z}_1 = \phantom{-}12.47 - 5.27z_1 + 40.00z_2$ 
 $\phantom{\dot{z}_1 =}+0.19z_1z_2 - 0.64z_1^2 - 0.93e^{z_1}$ 
 $\dot{z}_2 = -6.91 - 0.65z_1^2$ |

Table 4: Equation discovery for L-V system.

Table 3 and 4 show the equation discovery results for the nonlinear pendulum and the Lotka-Volterra system. For each dataset, we apply three methods for equation discovery: 1) learning a symmetric

latent space with `LaLiGAN`, and training SINDy with the fixed latent space; 2) training SINDy in the original observation space; 3) training SINDy autoencoder to learn a latent space without symmetry and discover the equation. Unlike the experiment in the high-dimensional reaction-diffusion system, we include SINDy without autoencoder because the observation space is low-dimensional in each of these systems and there indeed exists a closed-form governing equation.

It can be observed that applying `LaLiGAN` with SINDy still leads to simple linear equations. For the Lotka-Volterra system, the equation also consists of constant terms because the latent space is not centered at the origin. On the other hand, SINDy almost recovers the ground truth equations in both tasks, with no additional or missing terms but only small numerical errors for the coefficients.

However, our method can still achieve better long-term forecasting accuracy with the discovered equation. Similar to section 5.4, given an initial input frame $x_0$, we obtain its latent representation $\hat{z}_0 = \phi(x_0)$ and predict the future $T$ timesteps by iteratively calculating $\hat{z}_{t+1} = \hat{z}_t + F(\hat{z}_t) \cdot \Delta t$, where $\dot{z} = F(z)$ denotes the discovered governing equation. Then, we map the representations back to the input space by $\hat{x}_t = \psi(\hat{z}_t)$ to get the prediction in observation space. Figure 9 shows the relative mean square error at different timesteps. The curve for Neural ODE in the Lotka-Volterra system is incomplete because the prediction goes to NaN after about 1000 steps. Generally, our method leads to the slowest error accumulation. By contrast, while SINDy managed to recover the almost correct equation, the small numerical error can still lead to a very large error after a certain time period.

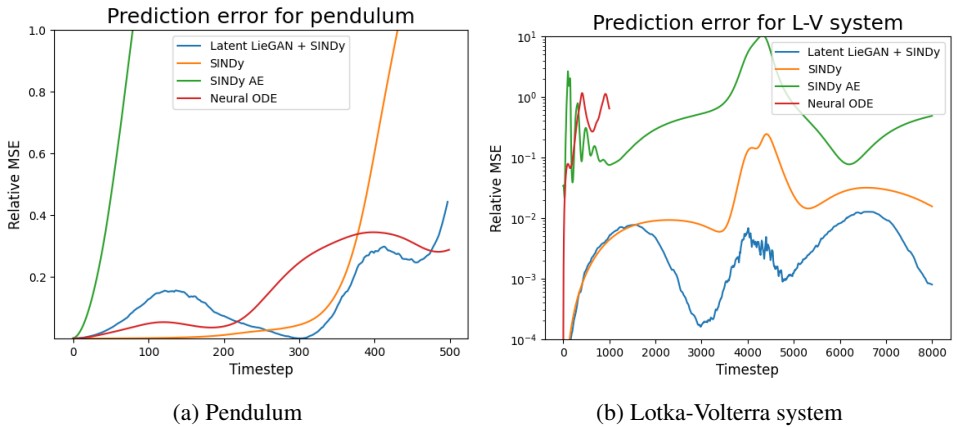

(a) Pendulum          (b) Lotka-Volterra system

Figure 9: Long-term prediction for nonlinear ODEs.

### A.3   MULTI-DIMENSIONAL LIE ALGEBRA

Our method learns a Lie algebra basis $\{L_i \in \mathbb{R}^{k \times k}\}_{i=1}^c$. The dimensionality of the Lie algebra, $c$, is a hyperparameter. In the previous experiments, we have set $c = 1$, meaning that `LaLiGAN` is only learning a one-dimensional Lie algebra at a time. Choosing a different $c$ allows us to discover multiple symmetries simultaneously in the latent space. We demonstrate this with the Lotka-Volterra equation experiment.

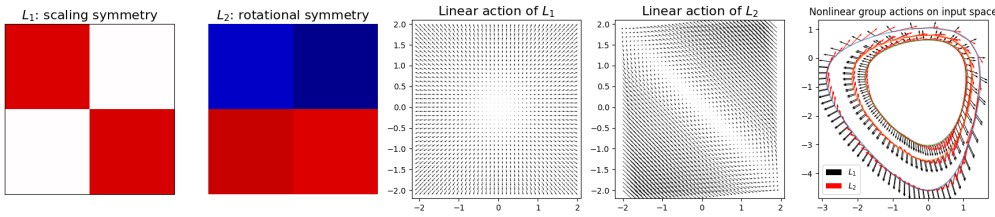

Figure 10: Discovered 2D Lie algebra and its actions on the latent space and the input space.

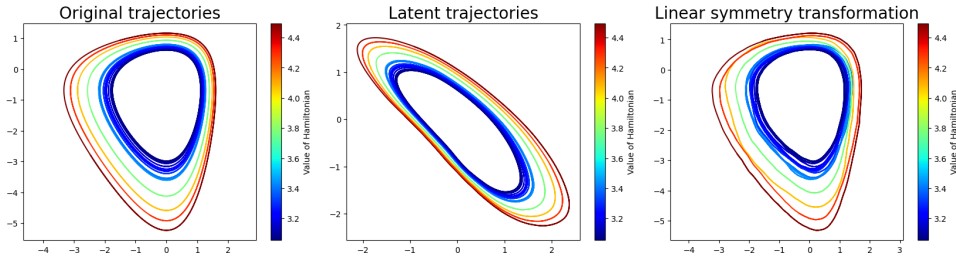

Figure 11: Left: Original trajectories of the Lotka-Volterra system. Middle: The trajectories mapped to the discovered latent space. Right: Original trajectories transformed by `LaLiGAN` with 2D Lie algebra.

In this experiment, we set the latent dimension to 2 and increase the Lie algebra dimension from 1 to 2. Figure 10 shows the discovered Lie algebra basis, $L_1$ and $L_2$. One can verify that $\{L_1, L_2\}$ forms a valid Lie algebra basis that is closed under the Lie bracket. The actions of $L_1$ and $L_2$ in the latent space can be visualized by the vector fields $(L_1z)^i\partial_i$ and $(L_2z)^i\partial_i$. It can be observed that $L_1$ corresponds to a scaling symmetry and $L_2$ corresponds to a rotational symmetry (up to a certain scaling and a tilt angle). The actions of of $L_1$ and $L_2$ in the input space can be visualized by the vector fields $(\frac{\partial\psi}{\partial z})^{ij}(L_1z)^j\partial_i$ and $(\frac{\partial\psi}{\partial z})^{ij}(L_2z)^j\partial_i$. The rightmost plot shows these vector fields evaluated on the original trajectories.

It is easier to interpret the meaning of these discovered symmetries by looking at the latent trajectories in Figure 11. The scaling symmetry $L_1$ changes the Hamiltonian of the system and indicates that the governing equation of the system $z_{t+1} = f(z_t)$ does not change with the Hamiltonian. The rotational symmetry $L_2$ is similar to the original experiment with only one-dimensional Lie algebra, which approximately takes one point to another within the same trajectory. Its representation differs from the previous one-dimensional experiment because the latent embeddings of the trajectories have also changed. Still, it can be interpreted as a time translation symmetry of the system.

## A.4 Learning SO(3) Equivariant Representation

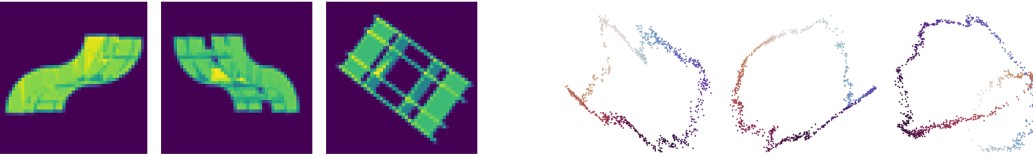

Figure 12: Left: the renderings of the object with three different orientations. Right: The object is rotated in three random axes from 0 to $2\pi$. The corresponding 2D images are embedded into a 3D latent space with `LaLiGAN`. For better visualization, the 3D latent representations are projected to 2D by PCA. The colors of the latent representations correspond to the rotation angles in $[0, 2\pi]$. The mapping to latent space $\phi$ is continuous with respect to the SO(3) rotations, and each latent trajectory of rotations around a particular axis roughly forms a circular manifold.

We consider a rotating bookshelf from ModelNet40 (Wu et al., 2015) and transform it in 3D through SO(3) rotations. The object is then rendered as a $48 \times 48$ image, which is the setting from Shakerinava et al. (2022). Figure 12 left displays the renderings of the object in three different orientations. The SO(3) action is nonlinear in the input space of 2D images. We use `LaLiGAN` to learn a latent space with 3 dimensions where the group action becomes linear.

Figure 12 right shows three latent trajectories. Each trajectory is obtained by rotating the object around a randomly selected axis in 3D space. The colors of the latent representations correspond to the rotation angles in $[0, 2\pi]$. The smooth transition of colors suggests that the mapping to latent space $\phi$ is continuous with respect to the SO(3) rotations. Also, each trajectory roughly forms a circular manifold.

We note that the trajectories are not in a perfect circular shape. For example, we observe that the latent representations overlap in some intervals. Concretely, given a particular rotation axis, let $x(\theta)$ denote the 2D rendering of the object with rotation angle $\theta$, and let $z(\theta) = \phi(x(\theta))$ denote its latent representation. In the 1st and 2nd latent trajectory shown in figure 12, it is observed that $z(\theta - \delta) \approx z(\theta + \delta)$ for some specific $\theta$ and small $\delta$'s. Also, in the 3rd trajectory, we have $z(\theta_1) \approx z(\theta_2)$ for some largely different $\theta_1$ and $\theta_2$. This can be caused by additional discrete symmetries in the object, where a transformation such as reflection or rotation up to $\pi$ leaves our view of the object unchanged. As our method is not provided with the group element associated with each object pose, it is unable to distinguish these identical inputs, so that they are mapped to the same location in the latent space and violate the overall circular structure. However, this kind of phenomenon does lead to an interesting question for future work: whether or not `LaLiGAN` can be extended to also discover these additional symmetries that are not caused by external transformations but lie in a real-world symmetric object itself.

## A.5 LEARNING THE LATENT TOROIDAL STRUCTURE OF FLATLAND

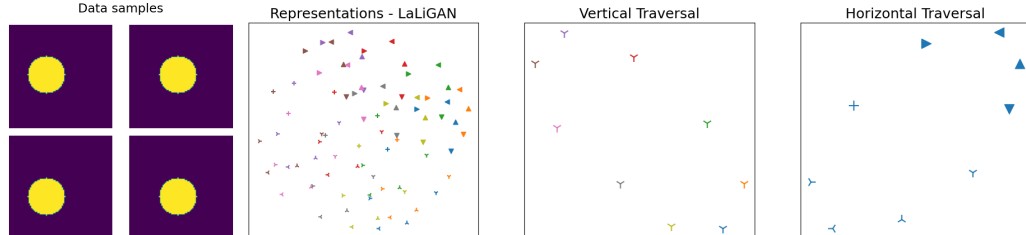

Figure 13: Left column: individual samples from the Flatland environment. Other columns: 2D projections of the 4D latent representations of equally spaced observations learned by `LaLiGAN`. Each marker style corresponds to a specific horizontal position of the ball. Each marker color corresponds to a specific vertical position. The latent space displays a toroidal structure, similar to the result in Figure 2 from Quessard et al. (2020).

Our method is related to symmetry-based disentangled representation learning (Caselles-Dupré et al., 2019; Quessard et al., 2020). For comparison, we consider a simple environment that is studied in these works, consisting of $84 \times 84$ pixel observations of a ball moving in a plane. The world is cyclic, meaning that the ball will appear at the top if it crosses the bottom boundary, and similarly for left/right. The observations are shown in Figure 13 (left).

Unlike Quessard et al. (2020) which considers sequences of observations $o$ and transformations $g$, $(o_0, g_0, o_1, g_1, ...)$, we are interested in discovering the *equivariance of a function*. Thus, we consider a function $o' = f(o)$ that simply translates the ball to the right and to the bottom by 15 pixels, respectively. An intuitive symmetry here is the cyclic translation equivariance along the two planar dimensions: if the input is translated by $g \in \mathrm{SO}(2) \times \mathrm{SO}(2)$, then the output will be translated by the same group element, i.e. $go' = f(go)$. In this experiment, we test whether `LaLiGAN` can discover a symmetry group of this function and a latent space where the group action becomes linear.

Following the setting in Quessard et al. (2020), we set the latent dimension to 4 and the search space of symmetries to $\mathrm{SO}(4)$. The discovered Lie algebra basis is

$$L_1 = \begin{bmatrix} 0 & 1.50 & -2.24 & 0 \\ -1.50 & 0 & 0 & 0 \\ 2.24 & 0 & 0 & 0 \\ 0 & 0 & 0 & 0 \end{bmatrix}, \ L_2 = \begin{bmatrix} 0 & 0 & 0 & 0 \\ 0 & 0 & 0 & -4.25 \\ 0 & 0 & 0 & -2.86 \\ 0 & 4.25 & 2.86 & 0 \end{bmatrix} \tag{6}$$

It can be verified that this basis forms a valid Lie algebra that is closed under the Lie bracket. It is a commutative Lie algebra that matches the structure of $\mathrm{SO}(2) \times \mathrm{SO}(2)$. Note that we do not enforce any disentanglement in the learning process, so the latent dimensions are entangled. Disentanglement can be explicitly enforced by decomposing the latent space as independent subspaces as described in Section 4.3, or promoted by encouraging the sparsity of the Lie algebra generators, but this is not the emphasis of this work.

The toroidal structure of the latent space can be verified from Figure 13. The visualization is obtained by projecting the 4D latent representations of equally spaced observations to 2D using Gaussian random projection. The marker colors and styles correspond to specific vertical and horizontal positions of the ball, respectively. It can be observed that all markers of a specific style, as well as all markers in a specific color, form a circular structure. For visual clarity, we also include two subsets: a vertical traversal along one column and a horizontal traversal along one row. This matches the result in Figure 2 from Quessard et al. (2020).

## A.6 Quantitative Evaluation of the Learned Symmetries

| Task | Symmetry | Equivariance error | Logit invariance error |
|---|---|---|---|
| Reaction-Diffusion | `LaLiGAN` | **1.02e-4** | **2.79e-3** |
| | LieGAN | - | 3.11e-2 |
| | SO(2) standard | 1.04e-4 | 2.84e-3 |
| Lotka-Volterra | `LaLiGAN` | **3.00e-2** | **5.21e-3** |
| | LieGAN | 8.44e-2 | 4.05e-1 |
| | SO(2) standard | 3.35e-2 | 5.68e-3 |
| Pendulum | `LaLiGAN` | **4.01e-3** | **5.33e-3** |
| | LieGAN | 6.30e-3 | 2.11e-2 |
| | SO(2) standard | 7.22e-3 | 1.57e-2 |

Table 5: Quantitative metrics for the learned symmetries on test datasets.

In this section, we introduce some metrics to evaluate the discovered symmetries quantitatively. Recall that the symmetries are the equivariances of a function, i.e. $f(gx) = gf(x)$. Thus, a straightforward metric would be the equivariance error defined as

$$EE_f = \mathbb{E}_{x,g}\|f(gx) - gf(x)\|^2. \tag{7}$$

Our symmetry discovery method is unsupervised and does not require fitting a function $f$. However, such a function can be fitted after discovering the symmetries, as is done in the dynamical system experiments. Concretely, the dataset consists of trajectories $\{x_{1:T}\}$, and the prediction function is $x_{t+1} = f(x_t)$. We use SINDy to learn symbolic equations $\dot{z} = h(z)$ (and therefore $z_{t+1} = H(z_t) = z_t + h(z_t)\Delta t$) in the latent space as shown in Table 2, 3 and 4. Then, $x_{t+1} = f(x_t) = (\psi \circ H \circ \phi)(x_t)$, where $\phi$ and $\psi$ are the learned encoder and decoder. Using this function $f$, we can evaluate the equivariance error of the learned symmetries on the test datasets. For comparison, we include the symmetry learned by linear LieGAN (Yang et al., 2023) in the input space (without autoencoder), where the function $f$ is the SINDy model trained in the input space (third row (SINDy) of Table 3 and 4). Note that this result is unavailable for the high-dimensional reaction-diffusion system because we did not train SINDy on its input space. Besides, we use the same autoencoder but replace the representation learned `LaLiGAN` with a standard representation of SO(2), i.e. $L = [0, -1; 1, 0] \in \mathbb{R}^{2\times2}$. Table 5 shows that `LaLiGAN` reaches the lowest equivariance errors on all of the three dynamical systems.

Another quantitative metric is inspired by the logit invariance introduced in Moskalev et al. (2023). For a classification task, we define the logit invariance error to measure the change of logits under group actions:

$$LI_f = \mathbb{E}_{x,g}\frac{1}{2}\|f(x) - f(gx)\|^2. \tag{8}$$

Here, the function $f$ outputs the logits for classification. In our setting, there is not necessarily such a classification function. However, we can utilize the learned discriminator in `LaLiGAN`, which effectively classifies between the original data distribution and the transformed distribution by the symmetry generator. A good symmetry should lead to a small difference between these two distributions. Therefore, we define the discriminator logit invariance error as follows:

$$DLI_f = \mathbb{E}_{v,g}\frac{1}{2}\|D(v) - D(gv)\|^2 \tag{9}$$

where $v = (x, y)$ are the data points sampled from the dataset. Table 5 shows that `LaLiGAN` has the lowest discriminator logit invariance error among the considered symmetries.

## A.7 LORENTZ SYMMETRY IN TOP TAGGING

We consider the Top Tagging dataset (Kasieczka et al., 2019), which is also studied in Yang et al. (2023). The task is a binary classification between top quark jets and the background signals. There are 2M observations in total, each consisting of the four-momentum of up to 200 particle jets.

This classification task is invariant to the restricted Lorentz group $SO^+(1,3)$. It is a 6-dimensional Lie group, including the spatial rotations around three axes and the boosts along three spatial directions.

The original dataset has a linear symmetry in the input space. To test whether LaLiGAN can learn nonlinear group actions, we transform the original inputs to a high-dimensional space and use it as the new input space for LaLiGAN. Concretely, we choose 4 spatial modes $\mathbf{u}_i \in \mathbb{R}^{128}$ given by Legendre polynomials and define $\mathbf{u} = \sum_{i=1}^{4} x_i \mathbf{u}_i$ where $\mathbf{x} = (x_1, x_2, x_3, x_4)$ is the 4-momentum from the original dataset.

In our experiment, we set the latent dimension to 4 and the Lie algebra dimension to 6. Figure 14 shows the discovered Lie algebra and its structure constants. Its representation is different from Figure 5 from Yang et al. (2023), because the latent representations obtained by the encoder are different from original 4-momentum inputs. However, we can compute the structure constants of this Lie algebra, which reveal its similar algebraic structure to the ground truth Lorentz algebra $\mathfrak{so}(1,3)$.

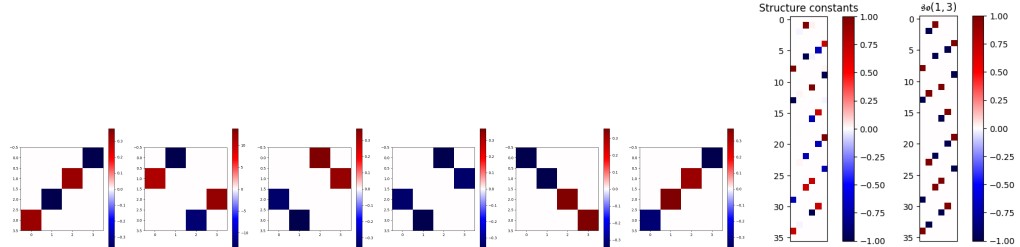

Figure 14: Left: Discovered 6-dimensional Lie algebra. Right: The structure constants of the discovered Lie algebra and the ground truth $\mathfrak{so}(1,3)$.

## B    PRELIMINARIES ON LIE GROUP REPRESENTATIONS

A Lie group is both a group and a differentiable manifold. We use Lie groups to describe continuous symmetry transformations. For example, the rotations in $\mathbb{R}^n$ form a Lie group $\mathrm{SO}(n)$; all rotations, translations and reflections in $\mathbb{R}^n$ form the Euclidean group $\mathrm{E}(n)$. We also referred to general linear group $\mathrm{GL}(n;\mathbb{R})$, which is the group of all $n \times n$ invertible matrices with real entries. As we only consider the field of real numbers in this work, we sometimes omit $\mathbb{R}$ and write $\mathrm{GL}(n)$ instead. We may also write $\mathrm{GL}(V)$, which is equivalent to $\mathrm{GL}(n,\mathbb{R})$ if $V = \mathbb{R}^n$ is a vector space.

The tangent vector space at the identity group element is called the Lie algebra of the Lie group $G$, denoted as $\mathfrak{g} = T_{\mathrm{id}}G$. The Lie algebra of general linear group $\mathrm{GL}(n,\mathbb{R})$ consists of all real-valued matrices of size $n \times n$. As Lie algebra is a vector space, we can use a basis $L_i \in \mathfrak{g}$ to describe any of its element as $A = \sum_{i=1}^c w_i L_i$, where $w_i \in \mathbb{R}$ and $c$ is the dimension of the vector space. Lie algebra can be interpreted as the space of infinitesimal transformations of the group. Group elements infinitesimally close to the identity can be written as $g = I + \sum_{i=1}^c w_i L_i$.

The exponential map $\exp : \mathfrak{g} \to G$ gives a mapping from the Lie algebra to the Lie group. For matrix Lie groups that we are considering, matrix exponential is such a map.

We are interested in how the data is transformed by group elements. Lie group, just like any other group, transforms the data from a vector space via a group action $\alpha : G \times V \to V$. If the action is linear, we call it a Lie group representation $\rho : G \to \mathrm{GL}(V)$, which acts on the vector space $V$ by matrix multiplication. Such a group representation induces a Lie algebra representation, $d\rho : \mathfrak{g} \to \mathfrak{gl}(V)$, which satisfies $\exp(d\rho(L)) = \rho(\exp(L)), \forall L \in \mathfrak{g}$.

Every matrix Lie group $G \leq \mathrm{GL}(n)$ has a standard representation, which is just the inclusion map of $G$ into $\mathrm{GL}(n)$. In our work, as we only consider these subgroups of general linear group, we learn the Lie group as its standard representation acting on $\mathbb{R}^n$ in the usual way. It is thus convenient to think of all group elements (and also Lie algebra elements) as $n \times n$ matrices, with the group operation given by matrix multiplication.

## C    UNIVERSAL APPROXIMATION OF NONLINEAR GROUP ACTIONS

### C.1    PROOFS

In this section, we provide theoretical justifications for the decomposition of nonlinear group actions introduced in section 4.1. We represent any nonlinear group action $\pi' : G \times V \to V$ as

$$\pi'(g, \cdot) = \psi \circ \pi(g) \circ \phi, \tag{10}$$

where $\phi : V \to Z$ and $\psi : Z \to V$ are functions parametrized by neural networks, and $\pi(g) : G \to \mathrm{GL}(k)$ is a group representation acting on the latent vector space $Z = \mathbb{R}^k$.

**Proposition C.1.** *If $\phi$ and $\psi$ are inverse of each other, then $\pi'(g, \cdot) = \psi \circ \pi(g) \circ \phi$ is a valid group action that satisfies identity and compatibility axioms.*

*Proof.* We have

$$\begin{aligned}
\pi'(g_2, \pi'(g_1, x)) &= \psi(\pi(g_2)\phi(g_1 \cdot x)) \\
&= \psi(\pi(g_2)\phi(\psi(\pi(g_1)\phi(x)))) \\
&= \psi(\pi(g_2)\pi(g_1)\phi(x)) \\
&= \psi(\pi(g_2 g_1)\phi(x)) \\
&= \pi'(g_2 g_1, x) \tag{11}
\end{aligned}$$
$$\pi'(e, x) = \psi(\phi(x)) = x \tag{12}$$

$\square$

The following theorem states that our proposed decomposition and neural network parametrization can approximate nonlinear group actions under certain conditions.

**Theorem C.2** (Universal Approximation of Nonlinear Group Action). *Let $G \leq \mathrm{GL}(k;\mathbb{R})$ be a compact Lie group that acts smoothly, freely and properly on $V = \mathbb{R}^k$ via a continuous group action $\pi' : G \times V \to V$. The group action, restricted to any bounded subset of the group, can be approximated by the decomposition $\pi'(g, \cdot) \approx \psi \circ \pi(g) \circ \phi$ if it admits a simply connected orbit space $V/G$, where $\psi$ and $\phi$ are fixed arbitrary-width neural networks with one hidden layer, and $\pi$ is a linear group representation.*

*Proof.* We establish our theorem as a corollary of the Universal Approximation Theorem (UAT) (Hornik et al., 1989), which states that any continuous function $f \in \mathcal{C}(\mathbb{R}^n, \mathbb{R}^m)$ can be approximated by a one-hidden-layer arbitrary-width neural network. The intuition of this proof is to explicitly construct the mappings between input and latent space and ensure their continuity so that we can use UAT to approximate them with neural nets.

The Quotient Manifold Theorem states that smooth, free, and proper group actions yield smooth manifolds as orbit spaces (Lee & Lee (2012), Theorem 21.10). More precisely, the orbit space $V/G$ has a unique smooth structure with a smooth submersion quotient map $s : V \to V/G$. Also, given that $V/G$ is simply connected, we can find a global continuous section $s' : V/G \to V$ s.t. $s' \circ s$ is identity restricted on $S = \mathrm{img}_{s'}(V/G)$. The global section can be constructed by:

1. Fix a base point $p \in V/G$ and choose a point $\tilde{p}$ in the pre-image of $p$ under $s$, i.e. $s(\tilde{p}) = p$.

2. For any other point $q \in V/G$, choose a path $\gamma$ in $V/G$ from $p$ to $q$.

3. As $V/G$ is simply connected, $V$ is a universal cover of $V/G$, so that any path $\gamma$ in $V/G$ can be uniquely lifted to a path $\tilde{\gamma}$ in $V$ which starts at $\tilde{p}$ and ends at $\tilde{q}$.

4. Define the section as $s' : V/G \to V, q \mapsto \tilde{q}$.

In addition, according to Whitney Embedding Theorem, the smooth manifold $V/G$ can be smoothly embedded in a higher-dimensional Euclidean space. Denote $t : V/G \to \mathbb{R}^p$ as one of the possible embeddings. We do not restrict the exact dimensionality of such an Euclidean space, as long as it enables us to represent any orbit with part of the latent space.

Before defining the mapping from input to latent, we finally note that as $G \leq \mathrm{GL}(k;\mathbb{R})$, we have a standard representation $\rho : G \to \mathbb{R}^{k \times k}$.

Now we define $\alpha : V \to \mathbb{R}^{k^2+p}, \pi'(g, s'(\tilde{v})) \mapsto \mathrm{concat}(\mathrm{vec}(\rho(g)), t(\tilde{v})), \forall \tilde{v} \in V/G, g \in G$, and we verify that this function is well defined.

First, $V = \{\pi'(g, s'(\tilde{v})) | \tilde{v} \in V/G, g \in G\}$, so that $\alpha(v)$ is defined for any $v \in V$.

Then, we need to make sure any $v \in V$ is written uniquely in the form of $v = \pi'(g)\tilde{v}$. $\forall \tilde{x}_i \neq \tilde{x}_j, g_i, g_j \in G, \pi'(g_i)\tilde{x}_i \neq \pi'(g_j)\tilde{x}_j$, because any two orbits never overlap in $V$.

Also, $\forall g_1, g_2 \in G, g_1 \neq g_2$, as $\pi'$ acts freely, we have $\pi'(g_1)\tilde{v} \neq \pi'(g_2)\tilde{v}$.

Next, we prove that $\alpha$ defined this way is also continuous. As the value of $\alpha$ is concatenated from two parts, it suffices to check the continuity for each component, i.e. $\alpha_1(\pi'(g, s'(\tilde{v}))) = \mathrm{vec}(\rho(g))$ and $\alpha_2(\pi'(g, s'(\tilde{v}))) = t(\tilde{v})$.

For any open set $t(\tilde{V}) \subset \mathbb{R}^p$, the continuity of $t$ and $s$ guarantees that the inverse image, $(t \circ s)^{-1} t(\tilde{V}) = s^{-1}(\tilde{V})$, is an open set. As $(s|_S)^{-1} = s'$, $s'(\tilde{V})$ is an open set. The $\alpha_2$ inverse image of $t(\tilde{V})$ is $\bigcup_{g \in G} \pi'(g, s'(\tilde{V}))$. Note that $\forall g \in G, \pi'(g^{-1}, \cdot) : V \to V$ is continuous, so that $\pi'(g, s'(\tilde{V}))$ is open. Therefore, the $\alpha_2$ inverse image of any open set $t(\tilde{V})$ is a union of open sets, which is also open, so that $\alpha_2$ is continuous.

Similarly, for any open set $\mathrm{vec}(\rho(U)) \in \mathbb{R}^k$, $U$ is an open set given the continuity of the standard representation $\rho$ and the vectorization operation. The $\alpha_1$ inverse image of $\mathrm{vec}(\rho(U))$ is $\bigcup_{s'(\tilde{v})} \pi'(U, s'(\tilde{v}))$. As the action of $G$ on $V$ is free, i.e. the stabilizer subgroup is trivial for all $v \in V$, we have $\pi'(\cdot, v) : G \to V$ is an injective continuous map, so that its image of an open set is still open. Thus, we conclude that $\alpha_1$ is also continuous.

According to the Universal Approximation Theorem, there exists a one-hidden-layer arbitrary-width neural network $\phi$ that approximates the continuous $\alpha$.

Then, we define $\pi(g) = (I_k \otimes \rho(g)) \oplus I_p$. For some $z_0 = (\text{vec}(\rho(g_0)), t(\tilde{v}_0))$ in the image of $\alpha$, we have

$$
\begin{aligned}
\pi(g)z_0 =& ((I_k \otimes \rho(g))\text{vec}(\rho(g_0)), t(\tilde{v}_0)) \\
=& (\text{vec}(\rho(g)\rho(g_0)), t(\tilde{v}_0)) \\
=& (\text{vec}(\rho(gg_0)), t(\tilde{v}_0))
\end{aligned}
$$

Finally, we define another mapping $\beta$ on $GZ = \bigcup_{g \in G, z \in Z} \pi(g)z$, where $Z$ is the image of $\alpha$, as $\beta : (\text{vec}(\rho(g)), t(\tilde{v})) \mapsto \pi'(g, s'(\tilde{v})), \forall \tilde{v} \in V/G, g \in G$. It is well-defined because $\text{vec} \circ \rho$ is injective on $G$, and also continuous because it is the inverse of $\alpha$. According to the Universal Approximation Theorem, there exists another neural network $\psi$ that approximates $\beta$.

Finally, defining $\alpha, \pi, \beta$ as above, for any $v = \pi'(g', s'(\tilde{v})) \in V$ and $g$ in any bounded subset of $G$, we have

$$
\begin{aligned}
\pi'(g, v) =& \pi'(g, \pi'(g', s'(\tilde{v}))) \\
=& \pi'(gg', s'(\tilde{v})) \\
=& \beta(\text{vec}(\rho(gg')), t(\tilde{v})) \\
=& (\beta \circ \pi(g))(\text{vec}(\rho(g')), t(\tilde{v})) \\
=& (\beta \circ \pi(g) \circ \alpha)(\pi'(g', s'(\tilde{v})) \\
=& (\beta \circ \pi(g) \circ \alpha)(v) \\
\approx& (\psi \circ \pi(g) \circ \phi)(v)
\end{aligned}
$$

The final step relies on the fact that the neural network approximator $\psi$ and the group representation $\pi(g)$ are Lipschitz continuous. Concretely, it requires $\|\psi(z_1) - \psi(z_2)\| \le K\|z_1 - z_2\|$, $\forall z_1, z_2$, for some positive constant $K$ and similarly for $\pi(g)$ as a function over $Z = \text{img}(\alpha)$. This is true for a one-layer neural network with ReLU activation, and also for $\pi(g)$ for any $g$ in a bounded subset of the group, because $\pi(g)$ is a bounded linear transformation.

Then, according to the UAT, for any $\epsilon > 0$, there exist neural networks $\psi$ and $\phi$ and positive constant $K$ s.t.

$$
\begin{aligned}
& \sup_{v \in V} \|(\psi \circ \pi(g) \circ \phi)(v) - (\beta \circ \pi(g) \circ \alpha)(v)\| \\
\le & \sup_{v \in V} \|(\psi \circ \pi(g) \circ \phi)(v) - (\psi \circ \pi(g) \circ \alpha)(v)\| + \|(\psi \circ \pi(g) \circ \alpha)(v) - (\beta \circ \pi(g) \circ \alpha)(v)\| \\
\le & \sup_{v \in V} K\|(\pi(g) \circ \phi)(v) - (\pi(g) \circ \alpha)(v)\| + \epsilon \\
\le & \sup_{v \in V} K^2 \|\phi(v) - \alpha(v)\| + \epsilon \\
\le & (K^2 + 1)\epsilon
\end{aligned}
$$

which translates to

$$
(\beta \circ \pi(g) \circ \alpha)(v) \approx (\psi \circ \pi(g) \circ \phi)(v)
$$

$\square$

## C.2 GROUP ACTION UNDER APPROXIMATE INVERSE

In practice, the networks $\phi$ and $\psi$ are trained with a reconstruction loss. As the loss is not strictly zero, they are only approximate but not perfect inverses of each other. As a result, the condition in Proposition C.1 cannot be strictly true. However, we can show empirically that when the reconstruction loss is reasonably close to zero, the decomposition in Proposition C.1 leads to an approximate group action. We use the reaction-diffusion system for demonstration.

A group action needs to satisfy the identity and compatibility axioms. We evaluate the error in terms of these axioms caused by the imperfect encoder and decoder networks. First, the error with respect to the identity axiom can be directly described by the reconstruction loss:

$$\text{err}_{\text{id}} = \mathbb{E}_x \|\pi'(e, x) - x\|^2 = \mathbb{E}_x \|\psi(\phi(x)) - x\|^2 = l_{\text{recon}}. \tag{13}$$

In the reaction-diffusion experiment, the test reconstruction loss is $2.58 \times 10^{-3}$, which indicates the autoencoder networks approximately satisfy the identity axiom.

Then, we consider the compatibility error. We sample a random group element $g$ from the generator and calculate $g^N$. Then, we apply $\pi'(g)^N = (\psi \circ \pi(g) \circ \phi)^N$ and $\pi'(g^N) = \psi \circ \pi(g^N) \circ \phi$ to the test dataset. The compatibility error is computed as

$$\text{err}_{\text{comp}} = \mathbb{E}_x \|\pi'(g)^N(x) - \pi'(g^N)(x)\|^2. \tag{14}$$

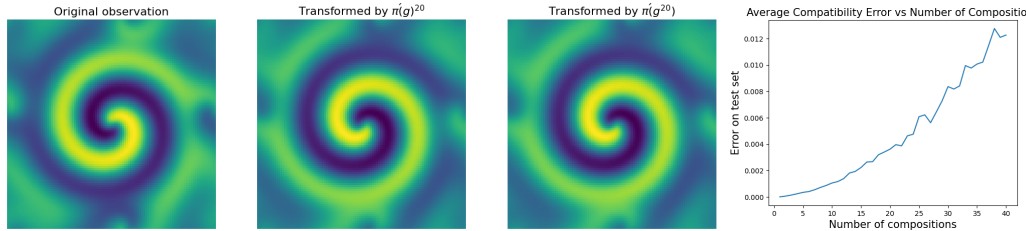

(a) An observation (left) transformed by $\pi'(g)^{20}$ (middle) and $\pi'(g^{20})$ (right)    (b) Compatibility error vs $N$

Figure 15: Group compatibility error caused by imperfect encoder and decoder networks.

Figure 15a shows a sample from test set transformed by $\pi'(g)^{20}$, that is, passed through the encoder, the linear representation $\pi(g)$ and the decoder for 20 times, and by $\pi'(g^{20})$, that is, passed through the autoencoder and the linear representation $\pi(g^{20})$ once. The two transformations have the same effect visually, which indicates that the autoencoder networks approximately satisfy the compatibility axiom. Further evidence is provided in Figure 15b, where we use the number of compositions $N \in [2, 40]$ and plot the growth of compatibility error with the increase of $N$. The error remains low ($\approx 1 \times 10^{-2}$) up to 40 times of group element composition.

## D  EXPERIMENT DETAILS

In this section, we provide the detailed hyperparameter settings and dataset generation procedure for the experiments.

### D.1  REACTION-DIFFUSION

We use the script from SINDy Autoencoder [1] to generate the dataset. We discretize the 2D space into a $100 \times 100$ grid, which leads to an input dimension of $10^4$. We simulate the system up to $T = 6000$ timesteps with step size $\Delta t = 0.05$. Then, we add random Gaussian noises with standard deviation $10^{-6}$ to each pixel and at each timestep. We use the timesteps $t \in [0, 4800)$ for training LaLiGAN and SINDy. For long-term forecasting, we use the timestep $t = 4800$ as the initial input frame and simulate up to 600 timesteps with each method. The simulations are then compared with the ground truth during $t \in [4800, 5400)$ to calculate the relative MSE.

We use MLPs with 5 hidden layers and 512 hidden units as the encoder, the decoder and the discriminator. We also use orthogonal parametrization for the final linear layer of the encoder, which is discussed in Section 4.3. The dimension of the Lie algebra in the LieGAN generator is set to one. We use a standard Gaussian as the distribution of the coefficient $w$ in the LieGAN generator.

For the 2D latent space symmetry discovery, we train for 150 epochs with batch size 64. The learning rates for the autoencoder, the generator and the discriminator are $0.0003, 0.001, 0.001$, respectively. The weights of the reconstruction loss and the GAN loss are set to $w_{\mathrm{recon}} = 1$ and $w_{\mathrm{GAN}} = 0.01$. As in LieGAN, we also include a regularization loss term $l_{\mathrm{reg}}$ for LieGAN generator, which pushes the Lie algebra basis away from zero, and the weight for the regularization is set to $w_{\mathrm{reg}} = 0.1$. We also apply sequential thresholding to the LieGAN generator parameters. Every 5 epochs, matrix entries with absolute values less than $0.01$ times the max absolute values across all entries are set to 0. For the 3D latent space, the settings are the same as above except that we train for 300 epochs.

For the equation discovery in the 2D and 3D latent spaces, we train SINDy for 300 epochs with batch size 64 and learning rate $0.001$. The SINDy function library contains polynomials up to the second order. We use $L_1$ regularization on the equation coefficients with weight $0.01$. We also apply sequential thresholding every 100 epochs, setting the coefficients less than $0.1$ to zero.

### D.2  NONLINEAR PENDULUM

We simulate the movement of nonlinear pendulum according to the governing equation, $\dot{q} = p, \dot{p} == -\sin(q)$. For training, we simulate 200 trajectories up to $T = 500$ timesteps with $\Delta t = 0.02$ with random initial conditions. For testing, we simulate another 20 trajectories. The initial conditions are sampled uniformly from $q_0 \in [-\pi, \pi]$ and $p_0 \in [-2.1, 2.1]$. Also, we ensure that $\mathcal{H} = \frac{1}{2}p^2 - \cos(q) < 0.99$, so that it does not lead to a circular movement.

We use MLPs with 5 hidden layers and 512 hidden units as the encoder, the decoder and the discriminator. We also use orthogonal parametrization for the final linear layer of the encoder and batch normalization before the transformation of the symmetry generator, as discussed in Section 4.3. The dimension of the Lie algebra in the LieGAN generator is set to one. We use a standard Gaussian as the distribution of the coefficient $w$ in the LieGAN generator.

We train for 70 epochs with batch size 256. The learning rate for the autoencoder, the generator and the discriminator are all $0.001$. The weights of the reconstruction loss and the GAN loss are set to $w_{\mathrm{recon}} = 1$ and $w_{\mathrm{GAN}} = 0.01$. The weight for the LieGAN regularization is set to $w_{\mathrm{reg}} = 0.02$. We also apply sequential thresholding to the LieGAN generator parameters. Every 5 epochs, matrix entries with absolute values less than $0.3$ times the max absolute values across all entries are set to 0.

For the equation discovery, we train SINDy for 30 epochs with batch size 64 and learning rate $0.001$. The SINDy function library contains polynomials up to the second order. We use $L_1$ regularization on the equation coefficients with weight $0.01$. We also apply sequential thresholding every 5 epochs, setting the coefficients less than $0.1$ to zero.

---

[1]https://github.com/kpchamp/SindyAutoencoders/tree/master/rd_solver

### D.3 Lotka-Volterra Equations

We simulate the Lotka-Volterra equations in its canonical form, $\dot{p} = a - be^q, \dot{q} = ce^p - d$, with $a = 2/3, b = 4/3, c = d = 1$. For training, we simulate 200 trajectories up to $T = 10000$ timesteps with $\Delta t = 0.002$ with random initial conditions. For testing, we simulate another 20 trajectories. The initial conditions are sampled by first sampling $x_0 = e^{p_0}$ and $y = e^{q_0}$ uniformly from $[0, 1]$ and then computing $p_0 = \log x_0$ and $q_0 = \log y_0$. Also, we ensure that the Hamiltonian of the system given by $\mathcal{H} = ce^p - dp + be^q - aq$ falls in the range of $[3, 4.5]$.

For all the experiments, we use MLPs with 5 hidden layers and 512 hidden units as the encoder, the decoder and the discriminator. We also use orthogonal parametrization for the final linear layer of the encoder and batch normalization before the transformation of the symmetry generator, as discussed in Section 4.3. The dimension of the Lie algebra in the LieGAN generator is set to one. We use a standard Gaussian as the distribution of the coefficient $w$ in the LieGAN generator.

We train for 30 epochs with batch size 8192. The learning rate for the autoencoder, the generator and the discriminator are all $0.001$. The weights of the reconstruction loss and the GAN loss are set to $w_{\text{recon}} = 1$ and $w_{\text{GAN}} = 0.01$. The weight for the LieGAN regularization is set to $w_{\text{reg}} = 0.01$. We also apply sequential thresholding to the LieGAN generator parameters. Every 5 epochs, matrix entries with absolute values less than $0.3$ times the max absolute values across all entries are set to 0.

For the equation discovery, we train SINDy for 5 epochs with batch size 64 and learning rate $0.001$. The SINDy function library contains polynomials up to the second order. We use $L_1$ regularization on the equation coefficients with weight $0.01$. We also apply sequential thresholding every epoch, setting the coefficients less than $0.1$ to zero.

### D.4 Double Bump

The signal length is set to 64, so that we have observations $x \in \mathbb{R}^{64}$. The rectangular and the triangular bump signals both have the length 16. For each sample, we randomly sample a shift $(\Delta_1, \Delta_2)$, where $\Delta_i$ is an integer in $[0, 64]$. The two bump signals are then cyclically shifted and superimposed. We sample 10000 signals for training and another 1000 for test.

We use a 1D convolution architecture for autoencoder. The encoder consists of three 1D convolution layers, with the numbers of input channels 1, 16, 32 and the final number of output channels 64, kernel size 3, stride 1 and padding 1, each followed by ReLU activation and a 1D max pooling layer with kernel size 2 and stride 2. The output of the final convolution is flattened and fed into an MLP with 2 hidden layers with 128 and 32 hidden units, and 4 output dimensions. The decoder structure is the reverse of the encoder structure, It consists of a 2-layer MLP with 32 and 128 hidden units, and 512 output dimensions. The MLP output is reshaped into 64 channels with size 8. Then three transposed convolution layers with output channels 32, 16, 1, kernel size 3, stride 2, input padding 1 and output padding 1 are applied. The final output passes through a sigmoid activation to ensure the output range is in $(0, 1)$. We use MLPs with 4 hidden layers and 128 hidden units as the discriminator. We also use orthogonal parametrization for the final linear layer of the encoder, as discussed in Section 4.3. The Lie algebra basis in the LieGAN generator is fixed to the standard representation of $\text{SO}(2) \times \text{SO}(2)$.

We train for 2000 epochs with batch size 64. The learning rate for the autoencoder and the discriminator are both $0.001$. The weights of the reconstruction loss and the GAN loss are set to $w_{\text{recon}} = 1$ and $w_{\text{GAN}} = 0.01$.

