# OpenReview forum: "Latent Space Symmetry Discovery"
_ICLR.cc/2024/Conference — Submitted to ICLR 2024_

### Official Review · Reviewer_BHEh · 2023-10-31

**Soundness:** 3 good
**Presentation:** 3 good
**Contribution:** 3 good
**Rating:** 6
**Confidence:** 3

**Summary:**

This paper extends LieGAN, which learns linear symmetries within data, to learn non-linear symmetries by integrating an autoencoder. The discovered symmetries/group transformations operate in the learned latent space (instead of in the input space as LieGAN) so that to be nonlinear.

Concretely, It decomposes the nonlinear group transformation into first encoding into the latent space, then linear transform, and lastly decoding back to the original space. It trains an autoencoder to ensure that the encoders and decoders are inverse of each other. It enforces the transformed data still to be in-distribution as the training dataset using a GAN loss.

The method is shown to learn rotation symmetric latent space for several dynamic systems. The learned latent space is shown helpful for equation discovery in one domain. The discovered equation is simpler and achieves better long-term prediction accuracy.

**Strengths:**

This paper discusses an important problem: discovering symmetries given the dataset. It is generally well-written and easy to read.

The method is intuitive, extending LieGAN with the latent space learned by autoencoders.

The experimental results show promising results in several dynamic systems, including one with a high-dimensional observation space.

**Weaknesses:**

I am mainly concerned with the practical applicability of this method:
* As discussed in the paper, the nonlinear-symmetric-discovery problem itself is ill-posed, and there are many meaningless "optimal" solutions to it due to the representation power of neural networks. This paper incorporates several patches to alleviate this issue, such as an orthogonal weight matrix in the final layer, and zero-mean of the latent features within an empirical batch. These regularization terms seem strong and hard-coded, and there is no theoretical understanding/analysis of them.
  - Are there metrics distinguishing the qualities of learned symmetries other than human interpretation?
  - Can the model, after applying all these regularization terms, learn all desired symmetries?
  - Are these regularization terms enough to rule out all meaningless solutions?
* Similarly, all learned/discovered symmetries in the experiments are rotation-based. Why is that? Is it related to the choice of the regularization? Can the model learn other symmetries in practice? For example, can it learn a nonlinear version of E(n)?
  - If the model can only learn rotation-based symmetries or rotation-based symmetries are enough with powerful neural encoders/decoders, why would we learn the symmetries in the latent space then?

Some other weakness includes
* The learned nonlinear symmetries are not that interpretable due to the neural encoder;
* It would be great to show an application area of the learned symmetries more than just the learned latent space.

**Questions:**

* How difficult is it to learn a meaningful nonlinear symmetry using this method in practice? Are there results showing this method learned symmetries other than those rotation-based? Are there results in domains other than the synthetic dynamic systems?
* Are there more ways to interpret/use the learned symmetries?

---

> ### Author Response · Authors · 2023-11-15
>
> ## Regarding the Regularizations
> Q1: As discussed in the paper, the nonlinear-symmetric-discovery problem itself is ill-posed, and there are many meaningless "optimal" solutions to it due to the representation power of neural networks. This paper incorporates several patches to alleviate this issue, such as an orthogonal weight matrix in the final layer, and zero-mean of the latent features within an empirical batch. These regularization terms seem strong and hard-coded, and there is no theoretical understanding/analysis of them.
>
> * The regularizations introduced in Sec 4.3 are necessary to address the failure modes in latent symmetry discovery. They are not hard-coded but well-motivated by the experiments. We provide more detailed discussion below.
>
> Q1.1: Are there metrics distinguishing the qualities of learned symmetries other than human interpretation?
>
> * **Yes**. A possible quantitative metric is the *equivariance error*: $E_{x,g}\|gf(x)-f(gx)\|^2$. Also, as Reviewer 5mQG has suggested, the *logit invariance* proposed by Moskalev et al (2023) are also related metrics. We have included these metrics in **Appendix A.6**.
>
> Q1.2: Can the model, after applying all these regularization terms, learn all desired symmetries?
>
> * **Yes**, these regularizations won’t remove any symmetries. First, the orthogonal parametrization would not cause any problem, because if we have an encoder with non-orthogonal final layer $W$, we can always do the Gram-Schmidt process to get an orthogonal weight $Q=PW$, which is effectively a change of basis in the latent space. Whatever linear symmetries in the original latent space should remain, possibly with a different group representation. Thus, in theory, an orthogonal final layer suffices to learn all desired symmetries. The zero-mean normalization would not affect most symmetries but does remove translation symmetry. However, the current group representation $v \mapsto \pi(g)v$ where $\pi(g)$ is a matrix cannot express translation of $v$ anyway, so the zero-mean normalization is compatible. To discover translation symmetries, though, we need to expand the search space to the affine group: $V \rtimes GL(V)$ and remove the normalization.
>
> Q1.3: Are these regularization terms enough to rule out all meaningless solutions?
>
> * We find that these regularizations resolve some common failure modes in the current experiments, but it is very difficult to enumerate all the unseen failures and rule them all out.
>
> ## Other Questions
> Q2: The learned nonlinear symmetries are not that interpretable due to the neural encoder.
> * The lack of interpretability comes naturally with the task of learning nonlinear group actions. It is often difficult to arrive at simple, closed-form formulas that describe all the nonlinear transformations considered, especially for high-dimensional systems. In our method, we managed to decouple the nonlinearity into neural components that do not rely on groups and group elements. Thus, we can analyze the group structure from the learned linear representations. This provides some degree of interpretability on the types of learned symmetries (e.g. rotation, scaling).
>
> Q3: It would be great to show an application area of the learned symmetries more than just the learned latent space.
> * See the following response to Q4.
>
> Q4: Are there more ways to interpret/use the learned symmetries?
> * **Yes**. Just as equivariant networks work on inputs with linear symmetries, after our method learns a symmetry with a linear action on the latent space, we can pass the latent embeddings to subsequent equivariant layers for downstream tasks. For example, we can make better use of the learned symmetry in the equation discovery algorithm by **enforcing the equivariance in SINDy regression**. Here is the idea in brief. SINDy learns a linear combination of basis functions: $\dot z=f(z)=WD(z)$, where $W$ is the learnable coefficients and $D$ is the function space basis. Mathematically we can show that if there is equivariance $g\dot z=f(gz)$, where $g$ is in a Lie group with $L$ in its Lie algebra, then $LWD(z)=W\frac{\partial D}{\partial z}Lz$. Similar to Equivariant MLP (Finzi et al, 2021), we can solve this linear constraint wrt trainable parameter $W$ and write $W=Q\beta$, where $Q$ is a constraint matrix and the vector $\beta$ becomes the actual trainable parameter. In this way, we are effectively using the learned symmetry as a constraint and compressing the search space of equations.

---

> ### Author Response · Authors · 2023-11-15
>
> Q5: How difficult is it to learn a meaningful nonlinear symmetry using this method in practice? Are there results showing this method learned symmetries other than those rotation-based? Are there results in domains other than the synthetic dynamic systems?
>
> * The success of symmetry discovery relies on proper choices for the hyperparameters, such as the dimensions of the Lie algebra and its representation, and the weights for the reconstruction loss and the GAN loss. Here are some principles for hyperparameter tuning. The principle for choosing the Lie algebra dimensionality is to start from a small value and gradually increase it until the generator cannot fool the discriminator, because it is required to produce more symmetries than there actually exist. Then, we choose a latent dimension that leads to a reasonably small reconstruction loss. This ensures that there is no information loss and the encoder and the decoder can be inverses of each other. The loss weights depend on different tasks, because the scales of the reconstruction loss vary.
> * **Yes, we provide some additional results of non-rotation-based symmetry and in domains other than synthetic dynamical systems.** The first example is the scaling symmetry in Lotka-Volterra system (**Appendix A.3**).
> Another example is **Top Tagging**, which is also used in LieGAN (Yang et al 2023). The task is a binary classification between top quark jets and the background signals. The original dataset is invariant to Lorentz transformations. To have a nonlinear symmetry, we apply a nonlinear transformation to the input: we choose 4 spatial modes $u_1,u_2,u_3,u_4 \in \mathbb R^{128}$ given by Legendre polynomials and define $x_{\text{new}}=\sum_{i=1}^4 x_iu_i$, where $x=(x_1,...,x_4)$ is the 4-momentum from the original dataset. (Note that this is really just an arbitrary choice of nonlinear transformation that does not have much physical meaning. Just a quick procedure to remove the linear symmetry and show our method’s ability to discover nonlinear group actions.) The discovered Lie algebra is [visualized HERE](https://postimg.cc/1gqmRN8m). Its representation is different from Figure 5 from Yang et al (2023), because the latent reps obtained by the encoder are different from original inputs. However, we can compute the structure constants of this Lie algebra ([visualized HERE](https://postimg.cc/gXbYxkhv)), which reveal its similar structure to the Lorentz algebra $\mathfrak{so}(1,3)$.
>
> ## References
> Moskalev A. et al. On genuine invariance learning without weight-tying. Topological, Algebraic and Geometric Learning Workshops 2023. – PMLR, 2023.
>
> Jianke Yang, Robin Walters, Nima Dehmamy, and Rose Yu. Generative adversarial symmetry discovery. International Conference on Machine Learning, 2023.
>
> Marc Finzi, Max Welling, and Andrew Gordon Wilson. A practical method for constructing equivariant multilayer perceptrons for arbitrary matrix groups. In International Conference on Machine Learning, pp. 3318–3328. PMLR, 2021.

---

> ### Comment · Reviewer_BHEh · 2023-11-15
> **A quick reply**
>
> I have read the rebuttals and appreciate the authors’ efforts in providing more experiments. Unfortunately, I couldn’t find convincing evidence to support me increasing the score. Most importantly, I couldn’t find evidence convincing me that this method is scalable and can be applied to discover useful unknown nonlinear symmetries in practice.
>
> In more detail, I understand the potential applications and ideas once an useful symmetry is learned. The questions were more about whether or not we could learn such useful symmetries. It is the same reason for me to find the theoretical issues “could” be less concerning in practice.
>
> Unfortunately, in my understanding, most experiments here use synthetic data or learn simple symmetries. For what it’s worth, meaningful symmetries in my understanding are, for example, discovering latent structures in PGM or even simple translation invariance in real world images. They may not be directly applicable but hopefully help clarify “useful” symmetries in practice from my perspective.
>
> Regarding the theoretical discussions, actually, the authors’ rebuttal on regularizations make me wonder if we need to “learn” lie transformations to achieve the same representation power as this method. Can we just fix a rotation-based latent symmetry space and ask encoders to map the input data points to corresponding positions in the latent space? I guess other reviewers would make better judgement on this.
>
> Regarding the novel discussions, I do not find this method too novel or non-novel. It is at least a new combination and I am most interested in the practical properties of it.

---

> ### Author Response · Authors · 2023-11-15
>
> We thank the reviewer for the timely feedback.
>
> In the additional experiments in Appendix A.3 and A.7, we have shown that our method can discover scaling and Lorentz symmetries. These are not simple symmetries and are highly meaningful in many scientific applications (Wang et al 2020, Bogatskiy et al 2020, Gong et al 2022).
>
> The top tagging experiment (Kasieczka et al 2019) in Appendix A.7 is a practical application in particle physics (Larkoski et al 2020, Karagiorgi et al 2022). And the Flatland experiment (Caselles-Dupré et al 2019, Quessard et al 2020) in Appendix A.3 suggested by Reviewer Vcu8 is done in an image environment. As pointed out by Reviewer 5mQG, our paper has already covered a wide range of tasks.
>
> We agree that real-world images and PGMs are good applications as well. (For images, translation symmetries are already well known, but it would be great if we could discover other less obvious symmetries.) These are certainly promising directions for future extension.
>
> ### References
>
> Wang, R., Walters, R., & Yu, R. (2020). Incorporating symmetry into deep dynamics models for improved generalization. *International Conference on Learning Representations*, 2021.
>
> Bogatskiy, A., Anderson, B., Offermann, J., Roussi, M., Miller, D., & Kondor, R. (2020, November). Lorentz group equivariant neural network for particle physics. In *International Conference on Machine Learning* (pp. 992-1002). PMLR.
>
> Gong, S., Meng, Q., Zhang, J., Qu, H., Li, C., Qian, S., ... & Liu, T. Y. (2022). An efficient Lorentz equivariant graph neural network for jet tagging. *Journal of High Energy Physics*, 2022(7), 1-22.
>
> Kasieczka, G., Plehn, T., Butter, A., Cranmer, K., Debnath, D., Dillon, B. M., ... & Varma, S. (2019). The machine learning landscape of top taggers. *SciPost Physics*, 7(1), 014.
>
> Larkoski, A. J., Moult, I., & Nachman, B. (2020). Jet substructure at the Large Hadron Collider: a review of recent advances in theory and machine learning. *Physics Reports*, 841, 1-63.
>
> Karagiorgi, G., Kasieczka, G., Kravitz, S., Nachman, B., & Shih, D. (2022). Machine learning in the search for new fundamental physics. *Nature Reviews Physics*, 4(6), 399-412.
>
> Caselles-Dupré, H., Garcia Ortiz, M., & Filliat, D. (2019). Symmetry-based disentangled representation learning requires interaction with environments. *Advances in Neural Information Processing Systems*, 32.
>
> Quessard, R., Barrett, T., & Clements, W. (2020). Learning disentangled representations and group structure of dynamical environments. *Advances in Neural Information Processing Systems*, 33, 19727-19737.

---

### Official Review · Reviewer_Vcu8 · 2023-11-01

**Soundness:** 1 poor
**Presentation:** 3 good
**Contribution:** 2 fair
**Rating:** 3
**Confidence:** 4

**Summary:**

This paper studies the problem of automatically discovering Lie group symmetries. To do so the paper focuses on non-linear group actions and attempts to discover linear representations in a latent space of an autoencoder. The overall method is termed Latent LieGAN and comes with a theory that attempts to show that the learned symmetry group is actually valid. In practical tests, LaLiGAN was able to recognize the inherent symmetry in high-dimensional data, creating a structured space that can be used for other tasks. The paper also showcases how LaLiGAN can be used to enhance equation discovery and make long-term predictions for different dynamic systems.

**Strengths:**

This paper studies an interesting problem, which started originally from the seminal work of Higgins et. al 2018. Since then, there has been a large body of work studying automatic symmetry discovery with various results. This paper adds to this body of work by using an adversarial approach which is easy to follow in this context. Unfortunately, I have a negative view of the originality and significance of this work as I outline in the next section but I will say the paper is generally well presented and shows a high degree of polish. The experimental results are quite toy and concocted but they make for good visuals and suggest there is merit in this approach to low dimensional problems.

**Weaknesses:**

I have several concerns regarding this paper to the point I am confused and question the validity of the entire endeavor. These might be my misunderstanding so I hope they can be clarified in the rebuttal. But as it stands I cannot endorse this paper for the following reasons.

1.) There is a strong emphasis on the non-linear group action aspect in this paper, but I believe this is a bit misguided. This is because the hallmark result of representation theory of Lie groups is that the Lie algebra connects the group to the vector space. Moreover, this can be described by matrices---hence linear representation---and you do not generally need non-linear representations. In practice, however, you can codify non-linear actions (e.g. rotations of 3D objects in a 2D image) and this is where you might want to learn a non-linear action. But I find the emphasis on the non-linear action exaggerated because LaLieGan learns a linear rep in the latent space anyways. I would suggest toning down these claims.

2.) Prop 4.1 seems to not apply to the setup that the authors consider. This is because the encoder and decoder map to a latent space of an autoencoder. This means that the latent dimension can be **lower** than the observation dimension. As a result, $\phi$ and $\psi$ cannot ever be inverses---i.e. bijective---because the information is lost. Thus, I have strong doubts about the value of the proposition. Moreover, many symmetry discovery methods already assume an autoencoder setup. The main difference is that they do not take an adversarial approach so this limits the novelty of the method. Finally, the paper learns approximate inverses anyways so there is no reason to guarantee that the learned representation is an exact Lie group.

3.) One of my biggest concerns is that the approach and results in this paper go against a relatively known result in Linear Symmetry Based Disentanglement by (Caselles-Dupré et. al 2019) who prove that symmetry discovery is impossible without interaction with the environment. This result is a symmetry-based analog to the result by Locatello et. al 2019. Thus I fear that the results in this paper are generally not true, and going beyond the toy datasets considered here might be impossible.

4.) I am confused as to why the authors do not compare with more standard baselines for symmetry discovery. Granted these works often assume knowledge of the group apriori but why this is not the correct test bed? For example $SO(N)$ is done in Fig 2 and 4 of Quessard et al 2020 (you even cite this paper) as well as the main experiment of Caselles-Dupré et. al 2019. Moreover, there has been a lot of development in Deep Delay auto-encoders that extend SINDy. In particular Deep Delay Autoencoders Bakkarji et. al 2023 is an appropriate baseline for the non-linear dynamical system discovery experiments. I encourage the authors to include this baseline as well.

**References**

Caselles-Dupré, Hugo, Michael Garcia Ortiz, and David Filliat. "Symmetry-based disentangled representation learning requires interaction with environments." Advances in Neural Information Processing Systems 32 (2019).

Locatello, Francesco, et al. "Challenging common assumptions in the unsupervised learning of disentangled representations." international conference on machine learning. PMLR, 2019.

Quessard, Robin, Thomas Barrett, and William Clements. "Learning disentangled representations and group structure of dynamical environments." Advances in Neural Information Processing Systems 33 (2020): 19727-19737.

Bakarji, Joseph, et al. "Discovering Governing Equations from Partial Measurements with Deep Delay Autoencoders. arXiv." arXiv preprint arXiv:2201.05136 (2022).

**Questions:**

See my questions in the weaknesses section.

---

> ### Author Response · Authors · 2023-11-15
> **Response to Weakness 1-2**
>
> ## Emphasis on the Nonlinear Group Action
> Q1: There is a strong emphasis on the non-linear group action aspect in this paper, but I believe this is a bit misguided. This is because the hallmark result of representation theory of Lie groups is that the Lie algebra connects the group to the vector space. Moreover, this can be described by matrices and you do not generally need non-linear representations. In practice, however, you can codify non-linear actions and this is where you might want to learn a non-linear action. But I find the emphasis on the non-linear action exaggerated because LaLieGan learns a linear rep in the latent space anyways. I would suggest toning down these claims.
> * We believe that discovering nonlinear group actions is an important aspect. Although representation theory is built on linear representations, in some scenarios, no linear action can describe the symmetries of the systems. This is illustrated in all our dynamical system experiments where LieGAN (Yang et al, 2023) fails. You mentioned that Lie algebra elements can be described by matrices, which is indeed an important fact. However, it is important to distinguish the group elements and their transformations on data space. The linear reps can describe the Lie algebra (and Lie group) elements, but they are not enough to describe their transformations on an arbitrary data manifold.
> * Therefore, we build the nonlinear action around a linear rep to enable symmetry discovery while also remaining compatible with the representation theory. As pointed out by the other reviewers, nonlinearity is an important next-step solution compared to existing methods e.g. LieGG (Moskalev et al, 2022) and LieGAN (Yang et al, 2023).
>
> ## Questions about Proposition 4.1
> Q2: Prop 4.1 seems to not apply to the setup that the authors consider because the encoder and decoder map to a latent space of an autoencoder. This means that the latent dimension can be lower than the observation dimension. As a result, phi and psi cannot ever be inverses because the information is lost. Thus, I have strong doubts about the value of the proposition. Moreover, many symmetry discovery methods already assume an autoencoder setup. The main difference is that they do not take an adversarial approach so this limits the novelty of the method. Finally, the paper learns approximate inverses anyways so there is no reason to guarantee that the learned representation is an exact Lie group.
> * In fact, Prop 4.1 **can** apply to our setup. In many tasks (e.g. the reaction-diffusion system in our paper), it is possible to encode the high-dimensional observations into a lower-dimensional latent space without loss of information. This is because the high-dimensional observations dwell on a low-dimensional manifold, and the intrinsic dimensions of these systems can be much lower. In such cases, the encoder $\phi$ is the inverse of the decoder $\psi$ restricted to the data manifold $\mathcal M \subset V$. Formally, Prop 4.1 is indeed problematic because it does not state this restriction to the data manifold. It should modified to: if $\phi|_{\mathcal M}$ and $\psi$ are inverse of each other, where $\mathcal M$ is the data manifold, then $\pi’(g, \cdot)$ is a valid group action. Thanks for pointing this out!
> * Regarding the novelty, our method does not require a priori knowledge of the symmetry group or additional information such as group element associated with each data point or interaction with environment. The latter is closely related to your next question and we provide further discussions there.
> * Finally, it is true that Prop 4.1 relies on an exact inverse condition. However, even when this condition is slightly violated in practice, it is still possible to learn a good group action. In Appendix C.2, we evaluate the validity of the learned group actions on the test dataset via the reconstruction loss and the compatibility error.

---

> > ### Comment · Reviewer_Vcu8 · 2023-11-20
> > **Re:Rebuttal**
> >
> > Thank you for your detailed response. I will have more to say before the end of this rebuttal. However, I still have problems with proposition 4.1. It is simply a vacuous statement. Of course, you can always restrict this to some unknown manifold in latent space and say that you have an inverse. We could say the same thing about VAEs and that the encoder and decoder are inverses given the natural image manifold (if the manifold hypothesis is true). Since you don't know $\mathcal{M}$ this statement doesn't add much value in my opinion. I would consider removing it from the main paper entirely or paint a more intuitive picture which is not a proposition.

---

> ### Author Response · Authors · 2023-11-15
> **Response to Weakness 3**
>
> ## Connection with SBDRL
> Q3: One of my biggest concerns is that the approach and results in this paper go against a relatively known result in SBDRL by (Caselles-Dupré et. al 2019) who prove that symmetry discovery is impossible without interaction with the environment.
>
> **No, our method does not contradict the results in SBDRL (Caselles-Dupré et al 2019),** because the task settings and assumptions are different:
> * By symmetry, we refer to the equivariance of a function $f$, i.e. $f(gx)=gf(x)$. For example, $f$ can be the evolution function in a dynamical system: $x_{t+1}=f(x_t)$. But it would be more helpful to think of $f$ as an arbitrary function, e.g. an image classifier / a time series forecaster, written as $y=f(x)$. Then, our dataset $\\{(x,y)\\}$ consists of input-output pairs of this function. We discover the equivariance of the function from the input-output pairs.
> * In Caselles-Dupré et al (2019) and Quessard et al (2020), the transitions from $o_t$ to $o_{t+1}$ rely on a group element $g$ instead. Their datasets are trajectories of $(o_0,g_0,o_1,g_1,...)$. They learn a map $f$ from observation $o\in W$ to latent $z\in Z$ that is equivariant between group actions on $W$ and $Z$.
>
> It should be noted that **many other works in symmetry discovery** belong to the first category. They **do not rely on interactions with the environment either.** (Zhou et al, 2021; Desai et al, 2022; Dehmamy et al, 2021; Yang et al, 2023)
>
> Despite the differences, SBDRL (Caselles-Dupré et al 2019), as well as its theoretical results, is surely related to our work. Next, we provide some discussions about the connection. For this part, we will follow the notations in the SBDRL paper.
>
> Theorem 1 in SBDRL states that there are multiple worlds equipped with different group actions on the state space $W$ that can produce the same dataset. Thus, the symmetry-based representation $(f, \cdot_Z)$ in one world is necessarily not equivariant in another world. This is indeed true, but it only means that there are multiple possible representations of symmetry, and our method can learn one of them.
>
> Using the hue change example (on page 4 immediately below Thm 1) from this paper, the succession of colors obtained by the right translation (i.e. $C_3$ group action on the state space) can be $(r, b, g, r)$ or $(r, g, b, r)$. For the first one, we can define equivariant representation as $f: r \mapsto 0, b \mapsto 1, g \mapsto 2$, and $hz = z + h\ \mathrm{mod}\ 3$ for $h \in \\{ 0, 1, 2 \\} = C_3$. This is indeed not a symmetry-based representation of the other world, but if we define the group action on the latent space as $hz = z - h\ \mathrm{mod}\ 3$ it becomes a valid representation.
>
> The setting of Thm 1 assumes there is a privileged group action attached to the environment and it is important to learn exactly that one. Thus it is problematic to have multiple $(W_i,\cdot_{W_i})$ which agree with training data and some of which are inconsistent with the learned $(f, \cdot_Z)$.
>
> In our case, however, we just wish to learn a group action which helps to understand the structure of the data manifold and can be used in downstream applications to reduce the dimensionality of the problem. Thus, each $(W_i,\cdot_{W_i})$ could be realized by a different $(f_i, \cdot_{Z_i})$, and we are satisfied with any of them. Thus, we believe that there is no contradiction.
>
> In retrospect, it might have been useful to specify the degree of non-uniqueness in the specification of our learning problem, i.e. given $(f, \cdot_Z)$, what can be done to produce a new $(f’, \cdot’_Z)$ which is just as good? Can we derive some equivalent relation $R$ and get the quotient space $\mathcal H / R$, where $\mathcal H \ni (f, \cdot_Z)$ denotes our hypothesis space? Answering these questions can reduce the complexity of our learning problem and may be a good direction for future work.

---

> ### Author Response · Authors · 2023-11-15
> **Response to Weakness 4 & References**
>
> ## Baselines
> Q4: Why not compare with more standard baselines for symmetry discovery. Granted these works often assume knowledge of the group apriori but why this is not the correct test bed? For example SO(N) is done in Fig 2 and 4 of Quessard et al 2020 as well as the main experiment of Caselles-Dupré et al 2019. I also encourage the authors to include Deep Delay Autoencoders Bakkarji et al 2023.
>
> **We didn’t compare because the settings of Quessard and Caselles-Dupré are different from ours, as discussed previously.** Assuming knowledge of the group is a minor difference. The main difference is that they consider a trajectory of observations and group transformations: $(o_0,g_0,o_1,g_1,...)$, while we aim to discover the equivariance of a function: $gf(x)=f(gx)$.
>
> But we can compare under a slightly modified setting, where we manually define such a function that is equivariant to cyclic translations. **We include the Flatland experiment as in Quessard and Caselles-Dupré in Appendix A.5, where we show that our method can learn similar 4D latent representations as in Figure 2 from Quessard.**
>
> Finally, regarding Delay SINDy autoencoder, there is no publicly available code. We tried to implement it by ourselves, but it did not yield good discovery results. There are many loss terms that are jointly optimized, and we find it difficult for all the reconstruction and consistency losses to converge simultaneously. We will not include this baseline for now. Any other suggestions on equation discovery baselines are welcome.
>
> We hope that our response can provide clarifications toward your questions. Please let us know if you have additional arguments against the paper or the rebuttal. If not, may we kindly request you to increase the score for our paper?
>
> ## References
> Artem Moskalev, Anna Sepliarskaia, Ivan Sosnovik, and Arnold Smeulders. Liegg: Studying learned lie group generators. Advances in Neural Information Processing Systems, 35:25212–25223, 2022.
>
> Jianke Yang, Robin Walters, Nima Dehmamy, and Rose Yu. Generative adversarial symmetry discovery. International Conference on Machine Learning, 2023.
>
> Hugo Caselles-Dupré, Michael Garcia Ortiz, and David Filliat. Symmetry-based disentangled representation learning requires interaction with environments. Advances in Neural Information Processing Systems, 32, 2019.
>
> Robin Quessard, Thomas Barrett, and William Clements. Learning disentangled representations and group structure of dynamical environments. Advances in Neural Information Processing Systems, 33:19727–19737, 2020.
>
> Allan Zhou, Tom Knowles, and Chelsea Finn. Meta-learning symmetries by reparameterization. International Conference on Learning Representations, 2021.
>
> Krish Desai, Benjamin Nachman, and Jesse Thaler. Symmetry discovery with deep learning. Physical Review D, 105(9):096031, 2022.
>
> Nima Dehmamy, Robin Walters, Yanchen Liu, Dashun Wang, and Rose Yu. Automatic symmetry discovery with lie algebra convolutional network. Advances in Neural Information Processing Systems, 34:2503–2515, 2021.

---

> > ### Comment · Reviewer_Vcu8 · 2023-11-23
> > **Re: Responses**
> >
> > Thank you for responding in detail to my review. Some points have been clarified, but others remain murky.
> > - The inclusion of deep delay autoencoders is quite important given that SINDy is an outdated baseline.
> > - Regarding the SBDL experiments, your claim is that the settings are different because one is focused on discovering the symmetries of a function while the other focus is on discovering symmetries of the world and matching that to the latent space. The authors claim that you only need to learn 1 world state which is equivariant. However, I am unconvinced that you are able to do that **exactly** not approximately. This is because the authors themselves have acknowledged that their encoders and decoders are not guaranteed to be inverses, so it is not a valid group action empirically---albeit close to one. Note that this paper is not about approximate equivariance, but exact equivariance of the function. This prevents us from making a rigorous claim on symmetry discovery.
> > - Another problematic point is the Discriminator. It is not clear to me why you need to sample group elements here. If you know that this is a matrix Lie group, you can make the Discriminator equivariant instead of doing approximate frame averaging---which is what you are doing.
> > - Finally, regarding the novelty of this work. Upon reflection, it seems this work is quite related to seminal works [1, 2]. They are effectively Wasserstein Autoencoders which is essentially what you have. However, the key difference seems to be that you parametrize the Lie algebra generators $\pi(g)$ as your latent map. Thus I view this work as less novel.
> >
> > Due to these factors, I am going to keep my current score.
> >
> > [1] Makhzani, A., Shlens, J., Jaitly, N., Goodfellow, I., & Frey, B. (2015). Adversarial autoencoders. arXiv preprint arXiv:1511.05644.
> >
> > [2] Tolstikhin, I., Bousquet, O., Gelly, S., & Schoelkopf, B. (2017). Wasserstein auto-encoders. arXiv preprint arXiv:1711.01558.

---

> > > ### Author Response · Authors · 2023-11-23
> > >
> > > We thank the reviewer for the feedback. We are glad our rebuttal provided some clarifications, including the important fact that our method is essentially different from Symmetry-based Disentanglement Representation Learning, and there is no contradiction between our work and the previous results.
> > >
> > > Regarding the new questions, we’d like to make some further comments.
> > >
> > > > The inclusion of Delay SINDy Autoencoder is important.
> > >
> > > We choose not to include Delay SINDy autoencoder because no official implementation is available and the result from our own implementation does not make sense. In fact, delay SINDy autoencoder aims at recovering the full equation from partial measurements. It’s input is one of the multiple state dimensions. When we have full states available as in our experiments, delay SINDy autoencoder does not have a special advantage over vanilla SINDy and may even require more efforts in hyperparameter tuning. The main idea we want to convey in the equation discovery part is that our method can regularize the latent space with symmetry and lead to more accurate long-term prediction, which is evident from the current results in Section 5.4 and Appendix A.1, A.2.
> > >
> > > > I am unconvinced that you are able to do that exactly not approximately. This is because the encoders and decoders are not guaranteed to be inverses, so it is not a valid group action empirically---albeit close to one. Note that this paper is not about approximate equivariance, but exact equivariance of the function. This prevents us from making a rigorous claim on symmetry discovery.
> > >
> > > Admittedly, the learned transformations do not strictly form a group action due to the approximate inverse. As we are learning symmetry from data with noise, the compatibility and identity axioms cannot be strictly guaranteed. But the error can be very low as we show in Appendix C.2 and A.6. See also our response to Reviewer **5mQG** (Question 3).
> > >
> > > > Another problematic point is the Discriminator. It is not clear to me why you need to sample group elements here. If you know that this is a matrix Lie group, you can make the Discriminator equivariant instead of doing approximate frame averaging---which is what you are doing.
> > >
> > > We don’t sample group elements in the discriminator, but in the generator. We conjecture that there is some misunderstanding about the central part of our method, so we restate the key idea here. The generator learns a distribution over general linear group. Through reparametrization, it samples group elements (as their standard representations) by sampling from a user-specified distribution on $\mathbb R^c$, $c$ being the Lie group dimension. The generator does **not** condition on $x$ or generate $x$. It produces group elements that transform $x$. We recommend going through Section 4.2 (in particular Eq. (3)) carefully.
> > >
> > > Also, the discriminator in our method does **not** do frame averaging. There is no frame map $F(x) \in 2^G$ in our method. This may be another confusion caused by the previous misunderstanding that the discriminator samples group elements.
> > >
> > > Then, using a equivariant discriminator does not serve our goal of symmetry discovery. First, we are learning the group, meaning it is changing constantly. But we need to build equivariant networks based on fixed and known groups. It is not something we can do differentiably on the fly. Even if we managed to build an equivariant discriminator, it would have prevented symmetry discovery. If it is equivariant to the transformations produced by the generator, then the generator output would have the same logit as the original data, so the generator cannot learn anything.
> > >
> > > In short, we believe these questions arise from some confusion about our method. Please explain more if we’ve misunderstood your point.
> > >
> > > > it seems this work is quite related to seminal works [1, 2]. They are effectively Wasserstein Autoencoders which is essentially what you have.
> > >
> > > We respectfully disagree that our method is essentially a Wasserstein autoencoder (WAE) or an adversarial autoencoder (AAE). For these two works, there is a prior distribution $p(z)$ we want to impose on the latent. This is done by measuring the discrepancy between $p(z)$ and either $\int q(z|x)dp(x)$ (WAE) or $q(z|x)$ using GAN (AAE). We do not have a prior distribution $p(z)$. Instead, we encourage the latent space to have some symmetries by including the GAN loss and the Lie generator.
> > >
> > > In the most general sense of relation, most autoencoder-based works include two loss terms: one for reconstruction and one for imposing some prior on the latent. It’s true that our method follows this paradigm, with the GAN loss imposing some prior about symmetry. But this paradigm relates most papers based on autoencoder. Thus we’d like to argue that such connection does not deny the novelty of our method.

---

> ### Author Response · Authors · 2023-11-20
> **A reminder**
>
> Dear reviewer,
>
> We have posted our response as above. Could you please let us know if you have additional arguments against the paper or the rebuttal? If there are additional issues, we'd like to have a further discussion. If not, may we kindly ask you to improve the support for our paper?

---

> ### Author Response · Authors · 2023-11-21
> **A quick reply regarding Proposition 4.1**
>
> By Prop 4.1, we want to emphasize that the group representation can be lifted to a nonlinear action if we have the bijective mapping. The inverse property is the condition but not the conclusion. It **motivates** us to train the model with reconstruction loss so it can achieve approximate inverse on the training data.
>
> Prop 4.1 is a formal statement to justify our method of learning the nonlinear actions. We can remove the ``Proposition'' header if you think it is more appropriate. But still we believe this statement should stay in the paper as it provides such justification.

---

### Official Review · Reviewer_5mQG · 2023-11-03

**Soundness:** 4 excellent
**Presentation:** 4 excellent
**Contribution:** 4 excellent
**Rating:** 8
**Confidence:** 5

**Summary:**

The paper considers the problem of symmetry estimation for the sake of better representation learning. The authors introduce Latent Lie GAN (LaLiGan) for learning non-linear symmetries in the input data. The paper highlights the fact that the problem has high importance for the field of representation learning. The authors demonstrate that there were many approaches for solving similar problems; however, the main focus was on linear symmetries, i.e., group representations. In contrast, the presented paper demonstrates that it is possible to learn non-linear symmetries in an adversarial manner.

**Strengths:**

- The paper is well-written. The flow is smooth and coherent. The paper presents good illustrations to help the reader understand the presented idea.
- The mathematical language is easy to follow, correct, and detailed when needed.
- The authors highlight the main contributions of the paper clearly.
- The presented method is clearly a next-step solution compared to approaches like LieGG or LieGAN.
- The experiments demonstrate that the proposed method can be applied in a wide range of tasks.

**Weaknesses:**

These are not significant weknesses. I would like to highlight the fact, that from the paper it seems like there are no natural limitations to the proposed method, which is however, not true. A straighforward explanation of situations when the method fails or can lead to an incorrect outcome will help

**Questions:**

I would like the authors to answer the following questions to make it easier to understand certain aspects of the method
- In Eq. 2 you learn transformations as $\sum_i\text{exp}[w_i L_i]$. How to choose the number of matrices L to be used in the method? I suppose the number of Lie algebra elements you parametrize will significantly affect the flexibility of the method in the latent space
- If I understood correctly, the proposed method works for compact groups only. The experiments demonstrate that the method can learn trajectories that are isomorphic to circles. How will the method behave on the data which has translation symmetry only? Will it fail? If so, the set of admissible symmetries seems more limited and should be highlighted
- in Proposition 4.1 you mention that $\psi$ and $\phi$ are inverse of each other. It is not correct, these functions are inverse to each other only on the input dataset. It raises the following question, how robust is the inverse property when you move away from the training dataset? How robust is the detected symmetry, when you move away from the training dataset? It reminds me of the following paper *Moskalev A. et al. On genuine invariance learning without weight-tying. Topological, Algebraic and Geometric Learning Workshops 2023. – PMLR, 2023*

---

> ### Author Response · Authors · 2023-11-15
>
> We anwer the reviewer's questions below.
>
> Q1: In Eq. 2 you learn transformations as exp(sum(w^iL_i).. How to choose the number of matrices L to be used in the method?
>
> * The number of matrices L, i.e. the dimension of Lie algebra, is a hyperparameter. A principle for choosing this hyperparameter is to start from a small value and gradually increase it until the generator cannot fool the discriminator, because it is required to produce more symmetries than there actually exist. In this way, we can discover the maximal symmetry. In the original experiments, we only considered one-dimensional Lie algebra. We provide an example of learning a two-dimensional Lie algebra for the Lotka-Volterra system in **Appendix A.3**.
>
> Q2: If I understood correctly, the proposed method works for compact groups only. The experiments demonstrate that the method can learn trajectories that are isomorphic to circles. How will the method behave on the data which has translation symmetry only? Will it fail? If so, the set of admissible symmetries seems more limited and should be highlighted
>
> * Our method **can** also discover non-compact groups. We provide an example in **Appendix A.3 (the scaling symmetry)**. Regarding translation symmetries, our current search space for symmetries is the general linear group which does not include translations. To discover translation symmetries, we need to expand the search space to affine group: $V \rtimes GL(V)$. We will state this limitation in the paper.
>
> Q3: in Proposition 4.1 you mention that phi & psi are inverse of each other. It is not correct, these functions are inverse to each other only on the input dataset. How robust is the inverse property when you move away from the training dataset? How robust is the detected symmetry, when you move away from the training dataset? It reminds me of the following paper Moskalev A. et al. On genuine invariance learning without weight-tying. Topological, Algebraic and Geometric Learning Workshops 2023. – PMLR, 2023
>
> * Prop 4.1 relies on the inverse condition. However, even when this condition is slightly violated in practice, it is still possible to learn a good group action. We show an example in Appendix C.2, where we evaluate the robustness of the learned group actions on the test dataset via the reconstruction loss and the compatibility error.
> * The paper from Moskalev proposes some useful metrics for measuring learned invariance. We can use our discriminator on the test set to evaluate the *logit invariance*. Also, another possible metric is *equivariance error*: $E_{x,g}\|gf(x)-f(gx)\|^2$. We have included these metrics in **Appendix A.6**. Thank you for pointing us to this paper!

---

> > ### Comment · Reviewer_5mQG · 2023-11-18
> > **Discussion**
> >
> > I would like to thank the authors of the paper for their answers.
> >
> > Could you explain, how you choose the dimensionality of $z$? Its deminsionality is the same as the dimensionality of the elements of the Lie Algebra. And I suppose that the accuracy with which you can learn the elements of the algebra depend on the dimensionality. Will yout method learn equally-accurate transformations if you choose $\text{dim}(z)=2,3,5,7,11$?

---

> > > ### Author Response · Authors · 2023-11-19
> > > **Discussion on latent dimensionality**
> > >
> > > We choose a latent dimension $k$ that is large enough to preserve the information in inputs. Practically, this can be done by initializing autoencoders with different latent dimensions and training them with reconstruction loss only. Then we choose the dimensionality that keeps the reconstruction loss below a reasonable threshold. In the experiments, we find $k=2$ suffices for the Reaction-Diffusion (R-D) system; the pendulum and Lotka-Volterra system are already low-dimensional so we use the same dimensionality as the input space. We have also studied your question about redundant dimension in **Sec 5.3**: using a 3D latent space for the R-D system. Our model still managed to learn a rotation symmetry in 3D.
> > >
> > > We can further evaluate the accuracy of the learned symmetry via the discriminator logit invariance metric (DLI) introduced in **Appendix A.6**. We train our model with different representation dimensions on R-D system and report the DLI error below (lower is better). The error increases with the redundant dimensions, suggesting that it is important to choose an appropriate latent dimension (large enough to allow accurate reconstruction, but not too large which complicates the learning of symmetry).
> > >
> > > | $k$                 | 2     | 3     | 5     | 7 | 11 |
> > > |---------------------|-------|-------|-------|---|----|
> > > | DLI $/ 10 ^ {-2}$ | $0.279$ | $0.056$ | $0.316$ |  $1.114$ |  $3.352$  |

---

### Author Response · Authors · 2023-11-15
**General Response to the Reviewers**

We thank the reviewers for their detailed and valuable feedback. We are encouraged that they find LaLiGAN to be a clearly motivated idea (R1) toward an important problem (R3). We are glad that they found our approach to be intuitive (R3), presented with coherent language and good illustrations (R1, R2), and demonstrated in a wide range of tasks (R1).

In response to the questions, we have added some experiments to the draft. Here we provide a summary of the new contents:
* Appendix A.3: Discovering a 2-dimensional Lie algebra from Lotka-Volterra system (additional scaling symmetry)
* Appendix A.5: Toroidal latent space of Flatland (comparison with works in disentangled representation learning)
* Appendix A.6: Quantitative metrics for the learned symmetries
* Appendix A.7: Discovering the Lorentz symmetry from Top Tagging dataset
* Appendix C.2: Evaluating the validity of group action under approximate inverse encoder & decoder

More detail can be found in our individual responses to each reviewer and in the revised draft. We have highlighted all changes from the original submission in blue text.

---

### Meta-Review · Area_Chair_4CGu · 2023-12-05

**Metareview:**

Although the diverging scores indicate that this could be a borderline paper, I feel that this paper has a substantal formal problem, in that  -- contrary to the the formal assumptions in the paper -- the learned encoder and decoder will (in general) not be inverses of each other due to a dimensionality mismach between the latent space and the input space. I consider this as highly problematic, because in small sample size applications I would expect that encoder and decoder will substantially differ in regions between the data points. In my opinion, this concern could not be addressed convincingly during the rebuttal and discussion phase (and I also doubt that there is a good solution to this problem).

**Justification For Why Not Higher Score:**

There is a severe formal problem, and I don't see any good strategy to overcome this problem.

**Justification For Why Not Lower Score:**

N/A

---

### Decision · Program_Chairs · 2024-01-16

Reject